# STREAMING VISUAL GEOMETRY TRANSFORMER

**Dong Zhuo**[*]   **Wenzhao Zheng**[*,†]   **Jiahe Guo**   **Yuqi Wu**   **Jie Zhou**   **Jiwen Lu**[✉]

Tsinghua University

https://wzzheng.net/StreamVGGT/

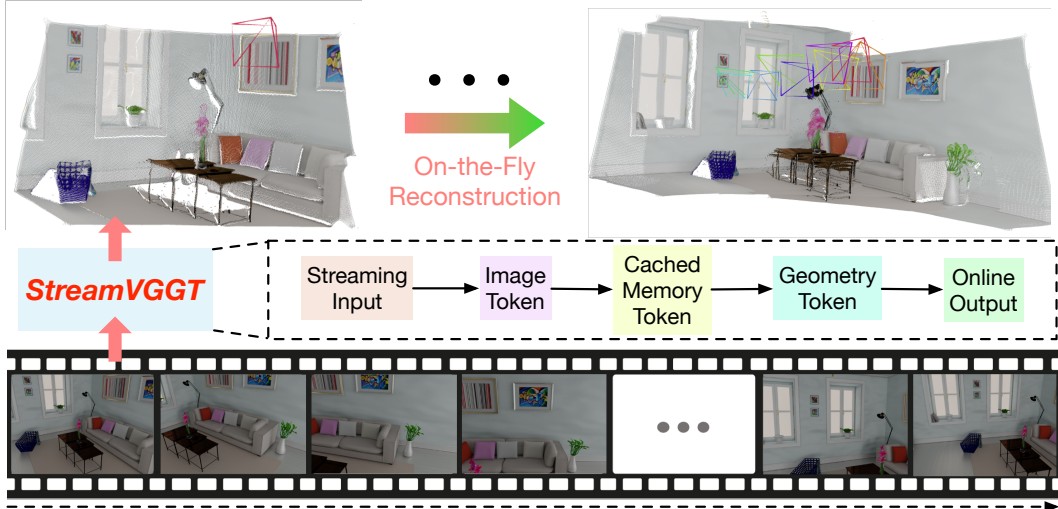

Figure 1: **Overview.** Unlike offline models that require reprocessing the entire sequence and reconstructing the entire scene upon receiving each new image, our StreamVGGT employs temporal causal attention and leverages cached token memory to support efficient incremental on-the-fly reconstruction, enabling interactive and low-latency online applications.

## ABSTRACT

Perceiving and reconstructing 3D geometry from videos is a fundamental yet challenging computer vision task. To facilitate interactive and low-latency applications, we propose a streaming visual geometry transformer that shares a similar philosophy with autoregressive large language models. We explore a simple and efficient design and employ a causal transformer architecture to process the input sequence in an online manner. We use temporal causal attention and cache the historical keys and values as implicit memory to enable efficient streaming long-term 3D reconstruction. This design can handle low-latency 3D reconstruction by incrementally integrating historical information while maintaining high-quality spatial consistency. For efficient training, we propose to distill knowledge from the dense bidirectional visual geometry grounded transformer (VGGT) to our causal model. For inference, our model supports the migration of optimized efficient attention operators (e.g., FlashAttention) from large language models. Extensive experiments on various 3D geometry perception benchmarks demonstrate that our model enhances inference speed in online scenarios while maintaining competitive performance, thereby facilitating scalable and interactive 3D vision systems.

## 1 INTRODUCTION

3D geometry reconstruction has long been a fundamental task in computer vision (Hartley & Zisserman, 2000; Özyeşil et al., 2017; Oliensis, 2000), which aims to estimate 3D geometry from a set of images. As a bridge between 2D images and the 3D world, it finds broad applications in diverse fields including autonomous driving (Huang et al., 2023; 2024), AR/VR (Zheng et al., 2024; Hong et al., 2024), and embodied robots (Wu et al., 2024; Wang et al., 2024b). With the development of

---

[*]Equal contributions. [†]Project leader. [✉] Corresponding Author.

embodied intelligence, on-the-fly 3D reconstruction from streaming inputs is increasingly demanded to enable online interactive visual systems, where latency and temporal consistency are crucial.

Conventional methods like Structure-from-Motion (SfM) (Snavely et al., 2006; Agarwal et al., 2011; Frahm et al., 2010; Wu, 2013; Schönberger & Frahm, 2016; Liu et al., 2025) and Multi-View Stereo (MVS) (Furukawa & Ponce, 2009; Gu et al., 2020) rely on explicit geometric constraints (Furukawa et al., 2015; Galliani et al., 2015) or global optimization (Niemeyer et al., 2020; Fu et al., 2022; Wei et al., 2021; Yariv et al., 2020), limiting scalability and speed. Recent learning-based approaches have shifted toward end-to-end frameworks that directly predict 3D structure from multi-view images. While pair-wise methods (Wang et al., 2024a; Leroy et al., 2024) have shown promising results by learning dense correspondences between image pairs, they utilize time-consuming post-processing steps for global alignment during multi-view reconstruction. Memory-augmented methods (Wang & Agapito, 2024; Wang et al., 2025b; Wu et al., 2025) maintain a memory pool to eliminate the need for post-processing. Additionally, their recursive memory update designs have the ability to address 3D reconstruction from videos. Nevertheless, error accumulation caused by causal architectures remains to be tackled. Fast3R (Yang et al., 2025) and VGGT (Wang et al., 2025a) circumvent iterative alignment by employing feed-forward architectures that enable global dense-view interactions. Despite achieving satisfactory performance, their dependence on global self-attention necessitates reprocessing the entire sequence at every step for frame-by-frame input and output scenarios. This offline paradigm precludes incremental reconstruction and diverges from the causal nature of human perception, limiting its practicality in streaming applications.

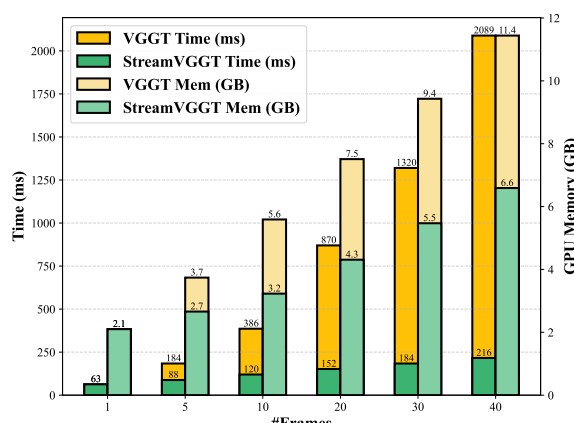

Figure 2: **Inference time and memory comparison** for the current frame of varying sequence lengths between StreamVGGT and VGGT for the online setting.

In this paper, we propose StreamVGGT, a causal transformer architecture specifically designed for efficient, low-latency streaming 3D visual geometry reconstruction, as shown in Figure 1. Unlike conventional offline frameworks that necessitate reprocessing the entire sequence with the arrival of each new frame, we introduce a temporal causal attention mechanism combined with an implicit historical-token memory module. This allows incremental processing of video frames, enabling progressive scene updates in an online streaming manner. StreamVGGT leverages the inherent sequential and causal nature of real-world video data and performs online streaming perception, aligning how humans observe a scene. Furthermore, to reduce training cost while retaining high accuracy, we adopt a distillation-based training strategy in which the densely connected, bidirectional VGGT (Wang et al., 2025a) serves as the teacher. By transferring its global geometric understanding, our causal student model achieves performance comparable to full-sequence inference with significantly fewer training resources and time. Distillation also unifies multi-task supervision through teacher-generated pseudo-GT, eliminating per-dataset engineering. In addition, the soft targets and confidence estimates of teacher act as effective regularizers, improving robustness and generalization. Equipped with FlashAttention-2 (Dao, 2023), our model supports fast inference for the current frame compared to VGGT, as shown in Figure 2. Experimental results demonstrate that our StreamVGGT reduces inference overhead in long-term sequences with only a slight performance trade-off, representing a crucial step toward low-latency responsive 3D vision systems.

## 2 RELATED WORK

**Conventional 3D Reconstruction.** 3D reconstruction, a fundamental task in computer vision (Hartley & Zisserman, 2000; Özyeşil et al., 2017; Oliensis, 2000), aims to recover the geometric structure of scenes from images or video sequences. Structure-from-Motion (SfM) (Snavely et al., 2006; Agarwal et al., 2011; Frahm et al., 2010; Wu, 2013; Schönberger & Frahm, 2016; Liu et al., 2025) reconstructs sparse 3D point clouds by matching image features across overlapping views and jointly refining camera poses and scene points through bundle adjustment (BA). A typical workflow comprises keypoint detection and description, feature matching with geometric verification, multi-view

initialization and incremental camera registration, triangulation, and a global bundle adjustment. Although highly accurate in static scenes, SfM is fragile in dynamic or texture-poor environments and, because of its computationally intensive offline optimization, cannot readily update in real time. Multi-View Stereo (MVS) (Furukawa et al., 2015; Galliani et al., 2015; Niemeyer et al., 2020; Fu et al., 2022; Wei et al., 2021; Yariv et al., 2020) exploits accurate camera poses to enforce photometric consistency across views, producing dense depth maps or point clouds that can be converted into high-resolution meshes or volumetric models. Neural Radiance Fields (NeRF) (Mildenhall et al., 2021) extend this idea by fitting the volume-rendering equation to learn a continuous radiance field, faithfully capturing fine-detail geometry and photometrically consistent appearance. Yet both MVS and NeRF depend on heavy offline optimisation, cannot update incrementally, and therefore provide limited low-latency capability. As a result, they are best employed as offline dense-reconstruction modules appended to an SfM pipeline rather than as online SLAM.

**Learning-Based 3D Reconstruction.** Building on the foundations of conventional 3D reconstruction, recent end-to-end, learning-based methods utilize neural networks to encode scene priors, markedly improving robustness and cross-dataset generalisation (Wang et al., 2024a; Leroy et al., 2024; Zhang et al., 2024; Smart et al., 2024; Fei et al., 2024; Dong et al., 2025). DUSt3R (Wang et al., 2024a) directly regresses view-consistent 3D point maps from just two RGB images without camera calibration, while its successor MASt3R (Leroy et al., 2024) introduces confidence-weighted losses to approximate metric scale. Recently, the feed-forward transformer-based architectures have emerged to enable dense-view interaction within a single pass (Yang et al., 2025; Wang et al., 2025a), delivering state-of-the-art accuracy in a few seconds. Fast3R (Yang et al., 2025) extends the pairwise DUSt3R idea to an N-view transformer equipped with memory-efficient Flash-Attention and parallel view fusion, allowing over 1000 images to be processed during inference. VGGT (Wang et al., 2025a) scales this philosophy to a 1.2B parameter visual geometry grounded transformer that jointly predicts camera intrinsics/extrinsics, dense depth, point maps, and 2D tracking features. However, its quadratic token-pair complexity and offline inference regime force complete re-encoding of every frame whenever a new image arrives. This heavy memory footprint and non-causal processing preclude streaming 3D reconstruction and undermine low-latency applications demanding instantaneous, frame-by-frame scene understanding.

**Streaming 3D Reconstruction.** low-latency streaming 3D reconstruction grows to be indispensable for autonomous driving, robotics, and AR/VR, where systems update scene geometry and camera pose on every frame with low latency (Wang & Agapito, 2024; Wang et al., 2025b; Wu et al., 2025). Spann3R (Wang & Agapito, 2024) augments a DUSt3R-style encoder with a token-addressable spatial memory, sustaining online point-map fusion but suffering drift on long or dynamic sequences due to its bounded memory. CUT3R (Wang et al., 2025b) introduces a recurrent transformer that jointly reads and writes a learnable scene state to output camera parameters, dense depth, and novel-view completions in real time. However, it degrades when extrapolating far from observed regions, and demands heavy computation for recursive training. Point3R (Wu et al., 2025) couples an explicit geometry-aligned spatial pointer memory with 3D hierarchical RoPE and an adaptive fusion mechanism, delivering low-drift online pose, depth, and point-map updates in real time. Instead of designing the memory mechanism to store past information, we follow the philosophy of large language models and employ a causal transformer to implicitly cache historical visual tokens.

## 3  PROPOSED APPROACH

### 3.1  STREAMING 3D GEOMETRY RECONSTRUCTION

Given a set of images $\{I_t\}_{t=1}^T$, where each frame $I_t \in \mathbb{R}^{3 \times H \times W}$, the generic 3D reconstruction pipeline can be written as:

$$F_t = \text{Encoder}(I_t), \ G_t = \text{Decoder}(F_t), \ (P_t, C_t) = \text{Head}(G_t), \tag{1}$$

where the encoder maps each input frame $I_t$ to a sequence of image tokens $F_t \in \mathbb{R}^{N \times C}$. A multi-view decoder then fuses cross-frame information, producing geometry tokens $G_t \in \mathbb{R}^{N \times C}$, and a MLP head predicts a point map $P_t \in \mathbb{R}^{3 \times H \times W}$ together with a per-pixel confidence map $C_t \in \mathbb{R}^{H \times W}$ from these geometry tokens.

This formulation provides a unifying paradigm for 3D reconstruction, but state-of-the-art systems differ in how the decoder aggregates multi-view information, which can be grouped into three cat-

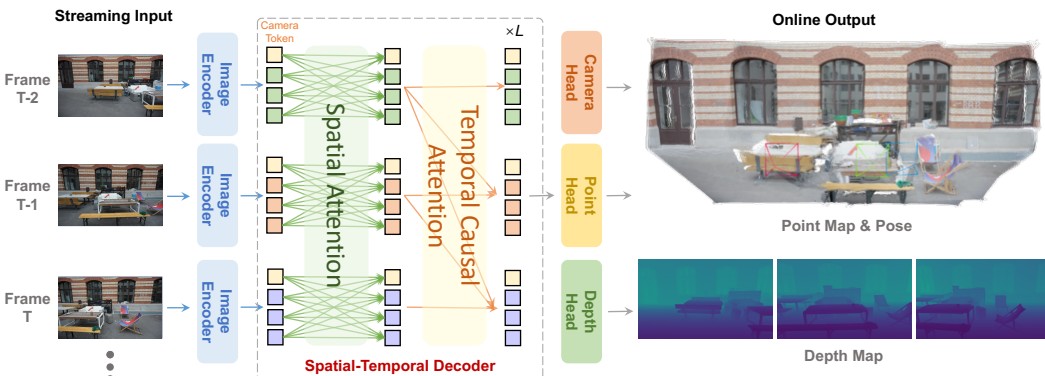

Figure 3: **Framework of StreamVGGT.** Our model consists of three main components: an image encoder, a spatio-temporal decoder, and multi-task prediction heads. During training, we utilize full-sequence inputs to provide the model with complete contextual information. To enforce temporal causality, we apply causal attention so the model can only attend to past frames at any given time step. This design encourages realistic temporal modeling suitable for streaming inference.

egories: pairwise, memory-augmented, and global interaction. Each framework introduced specialized modifications to the decoder and we summarize these designs as follows.

For pair-wise approaches like DUSt3R (Wang et al., 2024a) and MASt3R (Leroy et al., 2024), a two-branch cross-attention module jointly reasons over a reference/target pair, and no persistent state is kept beyond the current pair:

$$\{G_1, G_2\} = \text{Decoder}(\text{CrossAttn}(F_1, F_2)). \tag{2}$$

For memory-augmented approaches like Spann3R (Wang & Agapito, 2024) and CUT3R (Wang et al., 2025b), an external memory $M_t$ updated online enables global consistency without post-processing during long sequence inference:

$$G_t, M_t = \text{Decoder}(\text{CrossAttn}(F_t, M_{t-1})). \tag{3}$$

For global interaction approaches like Fast3R (Yang et al., 2025) and VGGT (Wang et al., 2025a), all frames attend to each other through all-to-all self-attention which achieves the highest accuracy at the cost of $\mathcal{O}(N^2)$ memory:

$$\{G_t\}_{t=1}^{T} = \text{Decoder}(\text{Global SelfAttn}(\{F_t\}_{t=1}^{T})). \tag{4}$$

Although global interaction approaches have achieved impressive reconstruction accuracy, their reliance on global self-attention mechanisms inherently limits their ability to process streaming inputs efficiently. To overcome this limitation, we utilize a causal transformer architecture to explicitly model the causal structure intrinsic for streaming data. Specifically, for our StreamVGGT:

$$\{G_t\}_{t=1}^{T} = \text{Decoder}(\text{Temporal SelfAttn}(\{F_t\}_{t=1}^{T})). \tag{5}$$

The temporal causal attention restricts each frame to attend only to itself and its predecessors, retaining rich context while reducing latency to $\mathcal{O}(N)$, which enables low-latency 3D perception.

## 3.2 CAUSAL ARCHITECTURE WITH CACHED MEMORY TOKEN

Building on the success of VGGT (Wang et al., 2025a), we propose a token-cached causal architecture consisting of three key components: image encoder, spatio-temporal decoder, and multi-task heads, as illustrated in Figure 3. We assume that the input images $I$ are provided in sequential order, and all output attributes $X$ are predicted frame by frame. This sequential input-output structure aligns with the causal perception logic observed in humans and is well-suited for low-latency 3D visual geometry reconstruction tasks, where spatial consistency and causality play a crucial role.

**Image Encoder.** We patchfy each input image $I_t$ into a set of $N$ image tokens $F_t \in \mathbb{R}^{N \times C}$ through DINO(Oquab et al., 2023). During the training stage, the image tokens of all frames are subsequently processed through our causal structure, alternating spatial and temporal attention layers.

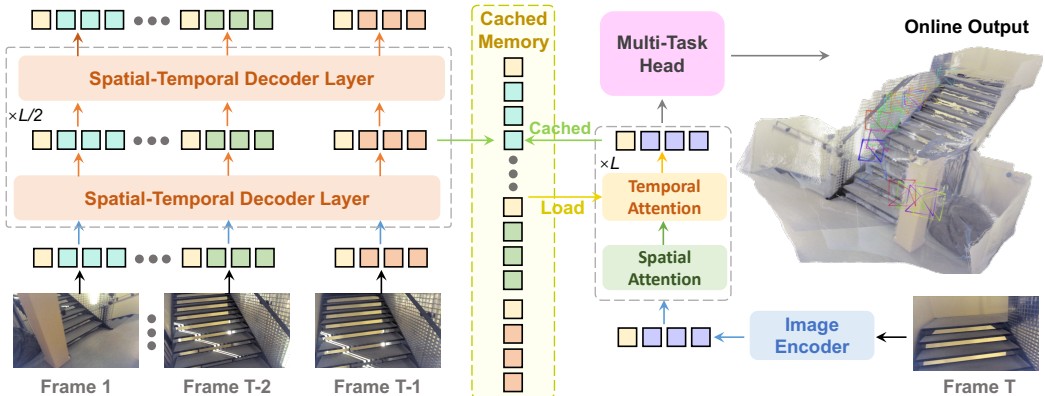

Figure 4: **Efficient inference of our model.** During streaming inference, we cache the historical keys and values as implicit memory to store information from past frames. This memory allows the model to efficiently reuse previously computed representations, avoiding redundant computation and enabling consistent contextual understanding across time. Our model then processes input incrementally and achieves performance that is comparable to full-sequence inference.

**Spatio-Temporal Decoder.** We introduce the spatio-temporal decoder by replacing all the global self-attention layers with temporal attention layers. In the standard global self-attention mechanism, each image token $F_t$ attends to all other tokens in the sequence, which can result in high computational costs when handling long sequences and is not well-suited for streaming 3D reconstruction tasks. In contrast, by using temporal attention, each token is restricted to attend only to the current and previous frames in the sequence, thereby respecting the inherent causal structure of the streaming inputs. This modification enables the model to maintain temporal consistency while significantly reducing the computational burden associated with global self-attention. In the training phase, we input all frames simultaneously, and the decoder generates geometry tokens $G_t$ based solely on the context from the historical and current frames. This causal self-attention mechanism lays the foundation for enabling streaming input inference with minimal latency.

**Cached Memory Token.** Unlike the training phase, where all frames are input simultaneously and processed with temporal attention, streaming 3D reconstruction requires the model to handle frame-by-frame input and perform incremental 3D reconstruction during inference. To address this, we introduce an implicit memory mechanism that caches historical token $M \in \mathbb{R}^{T \times N \times C}$ from previously processed frames, as shown in Figure 4. During inference, StreamVGGT performs cross attention between the cached memory tokens and the image tokens derived from the current frame:

$$G_T = \text{Decoder}(\text{CrossAttn}(F_T, \{M_t\}_{t=1}^{T-1})), \quad M_T = \text{TokenCachedMemory}(G_T). \tag{6}$$

This design enables the model to replicate the temporal causal attention behavior observed during training. Through experimental validation, we demonstrate that the model with cached memory token achieves performance comparable to that of full-sequence input inference. This confirms the effectiveness of our approach in maintaining high performance during streaming 3D reconstruction, making it well-suited for low-latency visual geometry reconstruction tasks.

**Multi-Task Heads.** Following the VGGT architecture, we utilize three distinct task heads, each dedicated to predicting key 3D attributes $X$ of the scene. These heads are responsible for estimating the camera pose $g_t \in \mathbb{R}^9$, point maps $P_t \in \mathbb{R}^{3 \times H \times W}$, depth maps $D_t \in \mathbb{R}^{H \times W}$, and point tracks $y_t \in \mathbb{R}^{2 \times M}$ from every single frame. Specifically, for each input image $I_t$, the image tokens $F_t$ produced by the image encoder are subsequently fed into the decoder to generate the geometry tokens $G_t$. The geometry tokens are then passed through specialized task heads that predict the desired 3D attributes. We also leverage a learnable camera token to mark the first frame as the global reference, so all subsequent frames are incrementally aligned within this shared coordinate system, enabling consistent, streaming 3D reconstruction without post-processing.

**Camera Head.** This head is responsible for predicting the intrinsic and extrinsic camera parameters. The output consists of the translation vector, rotation quaternion, and field of view (FoV), which together describe the pose of the camera in 3D space. The Camera Head uses self-attention layers to refine the camera parameters for each input frame.

**Geometry Head.** This head generates both the point map and depth map for each frame. Additionally, it outputs confidence maps (Kendall & Cipolla, 2016; Novotny et al., 2018) for both the point and depth predictions, indicating the certainty of the model in various regions. This head also produces dense tracking features, which are passed to the Track Head for point tracking across frames. Specifically, it leverages a DPT layer (Ranftl et al., 2021) to convert geometry tokens into dense feature maps, which are then processed by convolutional layers to produce the final point and depth maps with their corresponding confidence maps.

## 3.3 DISTILLATION-BASED TRAINING

**Knowledge Distillation.** To efficiently train our causal streaming architecture under limited computational resources, we employ a knowledge distillation (KD) strategy that transfers the geometric understanding of the bidirectional VGGT (Wang et al., 2025a) teacher into the causal student. Although causal transformers are naturally suited for low-latency inference, scalable training across datasets typically requires substantial annotation resources and engineering, which KD effectively avoids by enabling supervision without full ground-truth labels or per-dataset processing.

The VGGT teacher provides dense pseudo-labels for camera parameters, depth, point maps, and tracks, thereby unifying supervision across tasks and datasets. Its soft predictions and confidence estimates further offer uncertainty-aware guidance that stabilizes optimization and improves robustness, consistent with prior KD findings (Furlanello et al., 2018; Romero et al., 2014).

A key advantage of this strategy is the improvement in training efficiency. By inheriting the geometric priors and multi-view consistency of the teacher, the student achieves high-quality performance with far fewer GPU hours compared to training on full annotations. As shown in Section 4.6, under the same training time and computational budget, removing distillation leads to notable accuracy degradation, whereas the distilled student attains teacher-level performance while retaining the causal and streaming properties required for low-latency 3D perception.

**Training Loss.** We follow the loss design from VGGT (Wang et al., 2025a) but introduce a key difference: we use the output of a teacher model as pseudo-ground truth to supervise our causal student model. The teacher's soft targets and confidence estimates act as effective regularizers, improving robustness and generalization The loss function is:

$$L = L_{\text{camera}} + L_{\text{depth}} + L_{\text{pmap}}. \tag{7}$$

The camera loss $L_{\text{camera}}$ supervises the predicted camera parameters $\hat{g}_i$ by comparing them to the ground-truth camera parameters $g_i$. This is done using the Huber loss function:

$$L_{\text{camera}} = \sum_{i=1}^{N} \|\hat{g}_i - g_i\|_\epsilon, \tag{8}$$

where $\|\cdot\|_\epsilon$ represents the Huber loss function, which is robust to outliers in the data.

The depth loss $L_{\text{depth}}$ incorporates depth confidence, which weighs the discrepancy between the predicted depth $\hat{D}_i$ and the ground-truth depth $D_i$ with the predicted confidence map $\hat{\Sigma}_i^D$. The gradient-based term is applied to further refine the depth estimation, which is commonly used in monocular depth estimation tasks. The final form of the depth loss is:

$$L_{\text{depth}} = \sum_{i=1}^{N} \|\hat{\Sigma}_i^D \odot (\hat{D}_i - D_i)\| + \|\hat{\Sigma}_i^D \odot (\nabla \hat{D}_i - \nabla D_i)\| - \alpha \log \hat{\Sigma}_i^D, \tag{9}$$

where $\odot$ denotes the element-wise product, and $\nabla$ represents the gradient.

The point map loss $L_{\text{pmap}}$ is defined similarly to the depth loss but for the 3D point map. It takes into account the point-map confidence $\Sigma_i^P$ and ensures that the predicted 3D points $\hat{P}_i$ match the ground-truth points $P_i$. The loss is formulated as:

$$L_{\text{pmap}} = \sum_{i=1}^{N} \|\Sigma_i^P \odot (\hat{P}_i - P_i)\| + \|\Sigma_i^P \odot (\nabla \hat{P}_i - \nabla P_i)\| - \alpha \log \Sigma_i^P. \tag{10}$$

Table 1: **Quantitative 3D reconstruction results on 7-Scenes and NRGBD datasets.**

| | | 7 scenes | | | | | | NRGBD | | | | | |
| | | Acc↓ | | Comp↓ | | NC↑ | | Acc↓ | | Comp↓ | | NC↑ | |
| **Method** | **Type** | Mean | Med. | Mean | Med. | Mean | Med. | Mean | Med. | Mean | Med. | Mean | Med. |
| DUSt3R-GA (Wang et al., 2024a) | Pair-wise | 0.146 | 0.077 | 0.181 | 0.067 | 0.736 | 0.839 | 0.144 | 0.019 | 0.154 | **0.018** | 0.870 | 0.982 |
| MASt3R-GA (Leroy et al., 2024) | Pair-wise | 0.185 | 0.081 | 0.180 | 0.069 | 0.701 | 0.792 | 0.085 | 0.033 | **0.063** | 0.028 | 0.794 | 0.928 |
| MonST3R-GA (Zhang et al., 2024) | Pair-wise | 0.248 | 0.185 | 0.266 | 0.167 | 0.672 | 0.759 | 0.272 | 0.114 | 0.287 | 0.110 | 0.758 | 0.843 |
| VGGT (Wang et al., 2025a) | Dense-view | **0.088** | **0.039** | **0.091** | **0.039** | **0.787** | **0.890** | 0.073 | **0.018** | 0.077 | 0.021 | **0.910** | **0.990** |
| Spann3R (Wang & Agapito, 2024) | Streaming | 0.298 | 0.226 | 0.205 | 0.112 | 0.650 | 0.730 | 0.416 | 0.323 | 0.417 | 0.285 | 0.684 | 0.789 |
| CUT3R (Wang et al., 2025b) | Streaming | **0.126** | **0.047** | 0.154 | **0.031** | 0.727 | 0.834 | 0.099 | **0.031** | 0.076 | **0.026** | 0.837 | 0.971 |
| **StreamVGGT** | Streaming | 0.129 | 0.056 | **0.115** | 0.041 | **0.751** | **0.865** | **0.084** | 0.044 | **0.074** | 0.041 | **0.861** | **0.986** |

Table 2: **Quantitative 3D reconstruction results on ETH3D dataset.**

| Method | Type | Acc.↓ | Comp.↓ | Overall↓ |
| --- | --- | --- | --- | --- |
| DUSt3R (Wang et al., 2024a) | Pair-wise | 1.167 | 0.842 | 1.005 |
| MASt3R (Leroy et al., 2024) | Pair-wise | 0.968 | 0.684 | 0.826 |
| VGGT (Wang et al., 2025a) | Dense-view | **0.928** | **0.443** | **0.686** |
| CUT3R (Wang et al., 2025b) | Streaming | 1.426 | 1.395 | 1.411 |
| **StreamVGGT** | Streaming | **0.609** | **0.545** | **0.577** |

Table 3: **Quantitative 4D reconstruction results on TUM-dynamics.**

| Method | Type | Acc Mean↓ | Acc Med.↓ | Comp Mean↓ | Comp Med.↓ | NC Mean↑ | NC Med.↑ |
| --- | --- | --- | --- | --- | --- | --- | --- |
| VGGT (Wang et al., 2025a) | Dense-view | **0.050** | **0.008** | **0.055** | **0.017** | **0.622** | **0.695** |
| CUT3R (Wang et al., 2025b) | Streaming | 0.105 | 0.012 | 0.060 | **0.007** | 0.582 | 0.624 |
| **StreamVGGT** | Streaming | **0.085** | **0.011** | **0.058** | **0.007** | **0.617** | **0.690** |

Our framework unifies batch training and frame-by-frame inference through a causal, memory-cached architecture. During training, the student model distills the teacher's soft targets, which serve as effective regularizers that improve robustness and generalization, while significantly reducing the overall training cost. During inference, the cached memory tokens allow the model to process streaming inputs incrementally, replicating the causal behavior observed during training without sacrificing accuracy. This design ensures both high training efficiency and low-latency inference, making our method particularly well-suited for streaming 3D reconstruction.

## 4 EXPERIMENTS

### 4.1 IMPLEMENTATION DETAILS

Our model architecture follows VGGT (Wang et al., 2025a) with L = 24 layers of temporal and spatial attention modules. We integrated FlashAttention-2 (Dao, 2023) to accelerate the inference. We initialized StreamVGGT by using pre-trained weights from VGGT and fine-tuned approximately 950 million parameters (excluding the frozen image backbone) for 10 epochs. We employed the AdamW optimizer and a hybrid schedule of linear warm-up (first 0.5 epochs) followed by cosine decay, reaching a peak learning rate of 1e-6. For each training iteration, we randomly sampled a batch of 10 frames from diverse training scenes. Following Point3R (Wu et al., 2025), we processed input images with variable aspect ratios while resizing the maximum edge length to 518 pixels. Training was conducted on 4 NVIDIA A800 GPUs for 7 days.

### 4.2 TRAINING DATASETS

Our StreamVGGT was fine-tuned on a curated multi-domain collection comprising 13 datasets: Co3Dv2 (Reizenstein et al., 2021), BlendMVS (Yao et al., 2020), ARKitScenes (Baruch et al., 2021), MegaDepth (Li & Snavely, 2018), WildRGB (Xia et al., 2024), ScanNet (Dai et al., 2017), HyperSim (Roberts et al., 2021), OmniObject3D (Wu et al., 2023), MVS-Synth (Huang et al., 2018), PointOdyssey (Zheng et al., 2023), Virtual KITTI (Cabon et al., 2020), Spring (Mehl et al., 2023), and Waymo (Sun et al., 2020). This collection covers diverse visual domains spanning indoor/outdoor environments and temporal scales and balances synthetic data and real-world captures.

### 4.3 3D RECONSTRUCTION

**Comparisons on the 7-scenes and NRGBD datasets.** Following CUT3R (Wang et al., 2025b), we evaluated the 3D reconstruction performance of StreamVGGT on 7-Scenes (Shotton et al., 2013) and NRGBD (Azinović et al., 2022). Accuracy (Acc), completeness (Comp), and normal-consistency

Table 4: **Single-Frame Depth Evaluation.**

| Method | Type | Sintel Abs Rel ↓ | Sintel δ<1.25 ↑ | Bonn Abs Rel ↓ | Bonn δ<1.25 ↑ | KITTI Abs Rel ↓ | KITTI δ<1.25 ↑ | NYU-v2 (Static) Abs Rel ↓ | NYU-v2 (Static) δ<1.25 ↑ |
|---|---|---|---|---|---|---|---|---|---|
| DUSt3R (Wang et al., 2024a) | Pair-wise | 0.424 | 58.7 | 0.141 | 82.5 | 0.112 | 86.3 | 0.080 | 90.7 |
| MASt3R (Leroy et al., 2024) | Pair-wise | 0.340 | 60.4 | 0.142 | 82.0 | 0.079 | **94.7** | 0.129 | 84.9 |
| MonST3R (Zhang et al., 2024) | Pair-wise | 0.358 | 54.8 | 0.076 | 93.9 | 0.100 | 89.3 | 0.102 | 88.0 |
| VGGT (Wang et al., 2025a) | Dense-view | **0.276** | **67.5** | **0.055** | **97.1** | **0.072** | 93.8 | **0.060** | 95.1 |
| Spann3R (Wang & Agapito, 2024) | Streaming | 0.470 | 53.9 | 0.118 | 85.9 | 0.128 | 84.6 | 0.122 | 84.9 |
| CUT3R (Wang et al., 2025b) | Streaming | 0.428 | 55.4 | 0.063 | 96.2 | 0.092 | 91.3 | 0.086 | 90.9 |
| Point3R (Wu et al., 2025) | Streaming | 0.395 | 56.8 | 0.061 | 95.4 | 0.087 | 93.7 | 0.079 | 92.0 |
| **StreamVGGT** | Streaming | **0.254** | **68.5** | **0.052** | **97.1** | **0.072** | **94.7** | **0.055** | **95.9** |

Table 5: **Video Depth Evaluation.**

| Method | Type | Sintel Abs Rel ↓ | Sintel δ<1.25 ↑ | BONN Abs Rel ↓ | BONN δ<1.25 ↑ | KITTI Abs Rel ↓ | KITTI δ<1.25 ↑ |
|---|---|---|---|---|---|---|---|
| DUSt3R-GA (Wang et al., 2024a) | Pair-wise | 0.656 | 45.2 | 0.155 | 83.3 | 0.144 | 81.3 |
| MASt3R-GA (Leroy et al., 2024) | Pair-wise | 0.641 | 43.9 | 0.252 | 70.1 | 0.183 | 74.5 |
| MonST3R-GA (Zhang et al., 2024) | Pair-wise | 0.378 | 55.8 | 0.067 | 96.3 | 0.168 | 74.4 |
| VGGT (Wang et al., 2025a) | Dense-view | **0.298** | **68.1** | **0.057** | 96.8 | **0.061** | **97.0** |
| Spann3R (Wang & Agapito, 2024) | Streaming | 0.622 | 42.6 | 0.144 | 81.3 | 0.198 | 73.7 |
| CUT3R (Wang et al., 2025b) | Streaming | 0.421 | 47.9 | 0.078 | 93.7 | **0.118** | **88.1** |
| Point3R (Wu et al., 2025) | Streaming | 0.452 | 48.9 | 0.060 | 96.0 | 0.136 | 84.2 |
| **StreamVGGT** | Streaming | **0.323** | **65.7** | **0.059** | **97.2** | 0.173 | 72.1 |

(NC) scores are reported using sparse inputs (3–5 frames per scene on 7-Scenes and 2–4 frames per scene on NRGBD). Table 1 show that StreamVGGT performs competitively against existing streaming methods and surpasses the state-of-the-art streaming model CUT3R.

**Comparisons on the ETH3D dataset.** Following VGGT (Wang et al., 2025a), we further evaluated the accuracy of the predicted point clouds on the ETH3D (Schops et al., 2017) dataset. For each scene, we randomly sampled 10 frames and report the results after discarding invalid points using the valid masks. Specifically, we measure accuracy (Acc), completeness (Comp), and overall quality (Chamfer distance) for 3D reconstruction. Table 2 shows that StreamVGGT surpasses DUSt3R (Wang et al., 2024a) and MASt3R (Leroy et al., 2024) and matches the performance of the state-of-the-art VGGT, despite relying solely on information from the current and past frames. Note that our causal design enables streaming 3D reconstruction and is more efficient than VGGT.

**Comparisons on the TUM-dynamics Dataset.** To evaluate reconstruction performance of our StreamVGGT under dynamic scenarios, we conduct experiments on TUM-dynamics (Sturm et al., 2012) dataset, which contains dynamic object. Specifically, 50 frames are sampled from each sequence for this assessment. Based on these results in Table 3, we believe our method demonstrates strong capability for streaming 4D reconstruction in dynamic scenarios.

## 4.4 SINGLE-FRAME AND VIDEO DEPTH ESTIMATION

**Single-Frame Depth Estimation.** Following MonST3R (Zhang et al., 2024), we evaluated single-frame depth estimation across four datasets: KITTI (Geiger et al., 2013), Sintel (Alnegheimish et al., 2022), Bonn (Palazzolo et al., 2019), and NYU-v2 (Silberman et al., 2012), encompassing both dynamic/static scenes and indoor/outdoor environments. To ensure unbiased evaluation of cross-domain generalization, these datasets were strictly excluded during training. Following DUSt3R (Wang et al., 2024a), we employ two principal metrics: Absolute Relative Error (Abs Rel) and the percentage of predictions within a 1.25 factor of ground truth depth ($\delta 1.25$). Table 4 shows that our method not only matches the overall best performers but also outperforms the current state-of-the-art streaming model on all datasets, showing its superiority for online depth estimation.

**Video Depth Estimation.** We conducted video depth estimation by assessing both the depth quality on a per-frame basis and the consistency of depth across frames. This was done by aligning the predicted depth maps with the ground truth using a per-sequence scale. We show the comparison of these methods in Table 5. Under aligned settings, our StreamVGGT exceeds CUT3R on both the Sintel and Bonn benchmarks and attains performance comparable to the offline VGGT.

Table 6: **Camera Pose Estimation Evaluation** on ScanNet, Sintel, and TUM-dynamics datasets.

| Method | Type | ScanNet | | | Sintel | | | TUM-dynamics | | |
|---|---|---|---|---|---|---|---|---|---|---|
| | | ATE ↓ | RPE trans ↓ | RPE rot ↓ | ATE ↓ | RPE trans ↓ | RPE rot ↓ | ATE ↓ | RPE trans ↓ | RPE rot ↓ |
| Robust-CVD (Kopf et al., 2021) | Pair-wise | 0.227 | 0.064 | 7.374 | 0.360 | 0.154 | 3.443 | 0.153 | 0.026 | 3.528 |
| CasualSAM (Zhang et al., 2022) | Pair-wise | 0.158 | 0.034 | 1.618 | 0.141 | **0.035** | 0.615 | 0.071 | **0.010** | 1.712 |
| DUSt3R-GA (Wang et al., 2024a) | Pair-wise | 0.081 | 0.028 | 0.784 | 0.417 | 0.250 | 5.796 | 0.083 | 0.017 | 3.567 |
| MASt3R-GA (Leroy et al., 2024) | Pair-wise | 0.078 | 0.020 | 0.475 | 0.185 | 0.060 | 1.496 | 0.038 | 0.012 | 0.448 |
| MonST3R-GA (Zhang et al., 2024) | Pair-wise | 0.077 | 0.018 | 0.529 | **0.111** | 0.044 | 0.869 | 0.098 | 0.019 | 0.935 |
| VGGT (Wang et al., 2025a) | Dense-view | **0.035** | **0.015** | **0.377** | 0.169 | 0.064 | **0.474** | **0.012** | **0.010** | **0.307** |
| Spann3R (Wang & Agapito, 2024) | Streaming | 0.096 | 0.023 | 0.661 | 0.329 | 0.110 | 4.471 | 0.056 | 0.021 | 0.591 |
| CUT3R (Wang et al., 2025b) | Streaming | 0.099 | 0.022 | 0.600 | **0.213** | 0.066 | 0.621 | 0.046 | 0.015 | 0.473 |
| Point3R (Wu et al., 2025) | Streaming | 0.106 | 0.035 | 1.946 | 0.351 | 0.128 | 1.822 | 0.075 | 0.029 | 0.642 |
| **StreamVGGT** | Streaming | **0.048** | **0.019** | 0.557 | 0.219 | 0.103 | 1.041 | 0.026 | 0.012 | 0.316 |

Table 7: **The comparison of inference-time and memory consumption for the online setting.**

| Method | 1 frame Acc↓ / ms↓ / GB↓ | 5 frames Acc↓ / ms↓ / GB↓ | 10 frames Acc↓ / ms↓ / GB↓ | 20 frames Acc↓ / ms↓ / GB↓ | 30 frames Acc↓ / ms↓ / GB↓ | 40 frames Acc↓ / ms↓ / GB↓ |
|---|---|---|---|---|---|---|
| Spann3R (Wang & Agapito, 2024) | 0.039 / 500 / 5.3 | 0.038 / 87 / 5.4 | 0.046 / 82 / 5.5 | 0.036 / 120 / 5.5 | 0.043 / 192 / 5.6 | 0.068 / 125 / 5.7 |
| CUT3R (Wang et al., 2025b) | 0.035 / 101 / 3.4 | 0.038 / 102 / 3.5 | 0.039 / 99 / 3.6 | 0.031 / 101 / 3.8 | 0.027 / 102 / 4.0 | 0.023 / 102 / 4.2 |
| **StreamVGGT** | 0.028 / 63 / 2.1 | 0.024 / 88 / 2.7 | 0.024 / 120 / 3.2 | 0.021 / 152 / 4.3 | 0.021 / 184 / 5.5 | 0.021 / 216 / 6.6 |

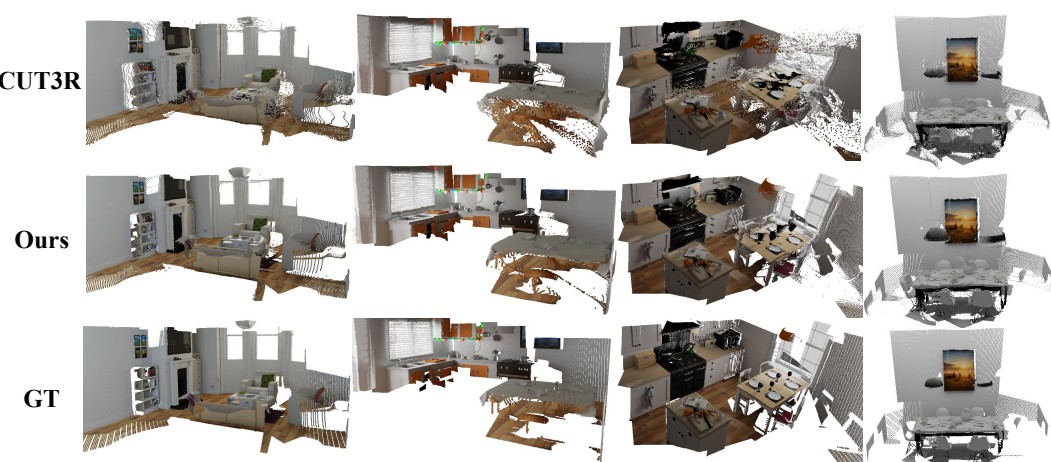

CUT3R

Ours

GT

Figure 5: **Qualitative comparisons with CUT3R.**

## 4.5 CAMERA POSE ESTIMATION

Table 6 shows the camera-pose estimation results on ScanNet (Dai et al., 2017), Sintel (Al-negheimish et al., 2022) and TUM dynamics (Sturm et al., 2012). Both Sintel and TUM-dynamics contain dynamic objects, which pose substantial challenges for conventional Structure-from-Motion (SfM) and SLAM methods. We evaluate performance by reporting the Absolute Translation Error (ATE), Relative Translation Error (RPE trans), and Relative Rotation Error (RPE rot), all computed after applying Sim(3) alignment with the ground truth trajectories. StreamVGGT delivers performance on par with the best existing methods while uniquely supporting camera-pose prediction from streaming inputs, highlighting its practicality for low-latency applications.

## 4.6 EXPERIMENTAL ANALYSIS

**Inference Time and Memory Consumption Analysis.** To evaluate the efficiency of our design in streaming tasks, we compare the inference latency for the final frame of sequences containing 1, 5, 10, 20, 30, and 40 frames among StreamVGGT and VGGT (Wang et al., 2025a). All experiments were conducted on a single NVIDIA A800 GPU. For StreamVGGT and VGGT, we employ FlashAttention-2 (Dao, 2023) with an image resolution of $518 \times 392$. The results of inference time and memory consumption are summarized in Figure 2. Additionally, we provide a comparison on inference consumption with CUT3R (Wang et al., 2025b) and Spann3R (Wang & Agapito, 2024). For a fair comparison, the input resolution for both CUT3R and Spann3R is set to $512 \times 384$. As shown in Table 7, our method exhibits a moderate increase in inference time and memory usage when processing longer sequences. This is mainly because the number of cached memory tokens grows with the sequence length. However, we reckon that there is a trade-off between spatial memory storage and performance, and StreamVGGT shows a good one for not very long sequences.

Table 8: **Comparison of pruning strategies for online inference on a 200-frame sequence.**

| Method | Acc↓ (Mean/Med.) | Comp↓ (Mean/Med.) | NC↑ (Mean/Med.) | Inference Time (Frame 200) | Peak Memory |
|---|---|---|---|---|---|
| Spann3R (Wang & Agapito, 2024) | 0.225 / 0.176 | 0.070 / 0.015 | 0.538 / 0.557 | 221.054 ms | **6.7 GB** |
| CUT3R (Wang et al., 2025b) | 0.165 / 0.107 | 0.030 / 0.012 | 0.551 / 0.577 | 102.533 ms | 7.6 GB |
| Ours No Pruning | 0.058 / 0.035 | 0.026 / 0.010 | 0.632 / 0.708 | 733.083 ms | 25.3 GB |
| Ours Window-based (50 frames) | **0.054 / 0.030** | **0.023 / 0.009** | **0.636 / 0.712** | 219.454 ms | 8.2 GB |
| Ours Window-based (100 frames) | 0.062 / 0.033 | 0.032 / 0.015 | 0.633 / 0.710 | 392.634 ms | 14.0 GB |
| Ours K-nearest-frames (K=50) | 0.157 / 0.114 | 0.104 / 0.011 | 0.605 / 0.670 | 220.109 ms | 8.4 GB |
| Ours K-nearest-frames (K=100) | **0.054 / 0.032** | 0.034 / 0.019 | 0.626 / 0.702 | 392.634 ms | 14.0 GB |

Table 9: **Effects of the distillation training strategy.**

| Method | Distillation | Attention | 7 scenes | | | | | | NRGBD | | | | | |
|---|---|---|---|---|---|---|---|---|---|---|---|---|---|---|
| | | | Acc↓ | | Comp↓ | | NC↑ | | Acc↓ | | Comp↓ | | NC↑ | |
| | | | Mean | Med. | Mean | Med. | Mean | Med. | Mean | Med. | Mean | Med. | Mean | Med. |
| VGGT (Wang et al., 2025a) | | Global | **0.088** | **0.039** | **0.091** | **0.039** | **0.787** | **0.890** | **0.073** | **0.018** | 0.077 | 0.021 | **0.910** | **0.990** |
| StreamVGGT (w/o KD) | | Causal | 0.202 | 0.102 | 0.168 | 0.064 | 0.718 | 0.825 | 0.189 | 0.088 | 0.206 | 0.096 | 0.816 | 0.945 |
| StreamVGGT (w/ KD) | ✓ | Causal | 0.129 | 0.056 | 0.115 | 0.041 | 0.751 | 0.865 | 0.084 | 0.044 | **0.074** | 0.041 | 0.861 | 0.986 |

Table 10: **Effects of the cached memory token mechanism for the online setting.**

| Method | Inference Time (at Frame 5) | Peak Memory |
|---|---|---|
| w/o FlashAttn & Cache | 1135.860 ms | 5.4 GB |
| w/ FlashAttn | 850.699 ms | **2.3 GB** |
| w/ FlashAttn & Cache | **88.193 ms** | 2.7 GB |

Still, StreamVGGT faces scalability challenge for very long sequences. Therefore, we introduce two strategies: (i) **windowed streaming**, which partitions the sequence into fixed-length chunks and then aligns the point clouds produced for each chunk using the predicted camera extrinsics, so that peak memory stays within a predefined budget; and (ii) **K-nearest-frames caching**, where each frame attends only to tokens from the most recent K frames. As demonstrated in Table 8, both approaches effectively bound memory usage and latency during inference on extended videos, while maintaining high accuracy. This offers a practical trade-off that effectively resolves computational concerns for long-sequence handling without significant degradation in results.

**Distillation Training Strategy.** To assess the effectiveness of our knowledge-distillation strategy, we evaluated three variants on the 7-Scenes (Shotton et al., 2013), NRGBD (Azinović et al., 2022), and ETH3D (Schops et al., 2017) datasets: (i) the global self-attention teacher VGGT (Wang et al., 2025a), (ii) StreamVGGT without KD, and (iii) StreamVGGT with KD. Table 9 shows that under the same training budget, the model trained without distillation exhibits higher reconstruction errors, indicating its limited ability to learn effective representations with constrained resources. In contrast, the distilled StreamVGGT achieves performance close to VGGT across all 3D reconstruction metrics while still operating within a low-latency, streaming architecture.

**Cached Memory Token Mechanism.** To evaluate the effect of our cached memory token mechanism and FlashAttention-2 (Dao, 2023) on inference speedup and memory reduction, we provide a ablation experiment in Table 10. The results clearly demonstrate that cached memory token mechanism reduces inference time and the FlashAttention-2 reduces the memory overhead.

**Visualizations**. StreamVGGT delivers photorealistic scene reconstructions with superior geometric fidelity, fewer outliers, and accuracy in complex environments, as shown in Figure 5.

## 5 CONCLUSION

In this paper, we have presented StreamVGGT, a causal transformer architecture for low-latency streaming 3D visual geometry reconstruction. By replacing global self-attention with causal temporal attention and introducing a cached token memory mechanism, StreamVGGT achieves incremental scene updates while preserving long-term spatial consistency. Extensive experiments demonstrate that StreamVGGT achieves comparable accuracy to the state-of-the-art offline model VGGT with small performance degradation, while surpassing current online state-of-the-art models across multiple tasks, including 3D reconstruction, single-frame depth estimation, and depth estimation. These advancements mark a critical step toward scalable, low-latency, streaming 3D vision systems for applications such as autonomous navigation and immersive AR/VR. Future work will explore dynamic scene adaptation and lightweight memory compression strategies to further bridge the gap between offline precision and mobile deployment requirements.

ACKNOWLEDGMENTS

This work was supported in part by the National Natural Science Foundation of China under Grant 62336004, Grant 62125603, Grant 62321005, and Grant 62576188, and in part by the Beijing Natural Science Foundation under Grant No. L247009.

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

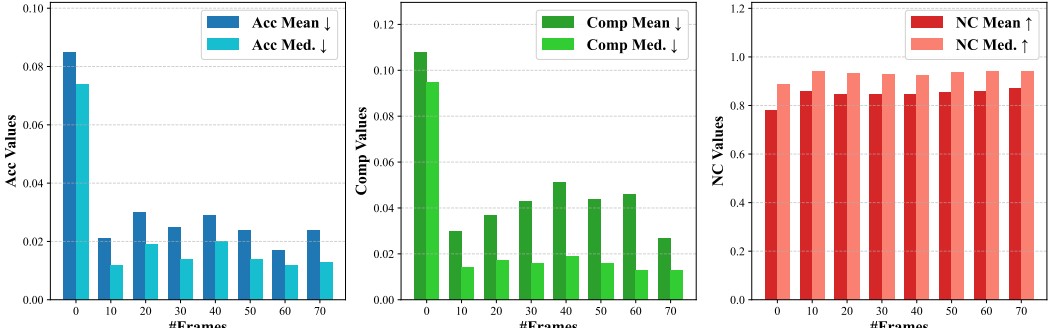

Figure 6: **Per-frame results on 7-Scenes (>70 frames).**

Table 11: **Camera pose estimation on 7-Scenes under loop-closure scenarios.**

| Method | Type | Avg. Translational Error (m)↓ | Avg. Rotational Error (°)↓ |
|---|---|---|---|
| CUT3R (Wang et al., 2025b) | Streaming | 0.0861 | 1.39 |
| **StreamVGGT** | Streaming | **0.0080** | **0.48** |

# A  APPENDIX

## A.1  RECONSTRUCTION PERFORMANCE IN COMPLEX SCENARIOS

It is important to evaluate the reconstruction performance under complex scenarios such as long-sequence or dynamic scenes. We now provide comprehensive quantitative results of our method.

**Per-frame Metrics on a Long Sequence.** To evaluate the temporal coherence of StreamVGGT on long-sequence. We conduct this experiment on the 7-Scenes (Shotton et al., 2013) dataset, each sequence includes over 70 frames. We report the per-frame evaluation metrics at every 10th frame as shown in Figure 6. The results demonstrate that our approach maintains robust temporal consistency.

**Loop Closure Analysis.** Specifically, for each sequence in the 7-Scenes (Shotton et al., 2013) dataset, we select the first 50 consecutive frames and extend them into a forward-and-reverse order to simulate loop closure scenarios. Then, we evaluate the camera pose errors between the first and last frames (the same frame) for each sequence to assess the performance under loop closure conditions. The results in Table 11 demonstrate that our model achieves lower drift and usable performance in loop closure scenarios.

**Quantitative Reconstruction Performance when the Video Stream is Interrupted.** It is important for streaming methods to maintain usable performance when the input video stream is interrupted for a period of time. As a result, we provide quantitative reconstruction results of our model on the 7-scenes (Shotton et al., 2013) dataset, where for each sequence we pick the first 50 frames and then remove the middle consecutive 25 frames to simulate the scenarios. The results in Table 12 demonstrate that our model can handle this challenging scenario better than the baseline CUT3R (Wang et al., 2025b).

**Quantitative Reconstruction Performance to Noisy Frames.** For noisy frames, we further corrupt inputs by randomly masking regions to simulate sensor noise. Table 13 shows reconstruction remains correct on the 7-scenes (Shotton et al., 2013) dataset, which proves the robustness of our StreamVGGT to noisy frames.

## A.2  MORE ANALYSIS

**Cached Memory Token.** To validate the effectiveness of the cached memory token, we evaluated StreamVGGT on the ETH3D (Schops et al., 2017) dataset under two settings: full-sequence input and streaming input, as shown in Table 14. "Time" denotes the inference latency of the final frame in a 40-frame stream. The results indicate that cached memory tokens not only accelerate streaming inference but also preserve temporal consistency and perceptual accuracy.

**More Visualizations.** Although some of our reconstruction results are presented in the main paper, we provide additional visualizations of the reconstruction outcomes of StreamVGGT and

Table 12: **Results on 7-Scenes when the video stream is interrupted.**

| Method | Type | Acc Mean↓ | Acc Med.↓ | Comp Mean↓ | Comp Med.↓ | NC Mean↑ | NC Med.↑ |
|---|---|---|---|---|---|---|---|
| CUT3R (Wang et al., 2025b) | Streaming | 0.042 | 0.024 | 0.037 | 0.016 | 0.702 | 0.808 |
| **StreamVGGT** | Streaming | **0.036** | **0.020** | **0.033** | **0.012** | **0.724** | **0.833** |

Table 13: **The robustness performance under noisy frames.**

| Method | Acc Mean↓ | Acc Med.↓ | Comp Mean↓ | Comp Med.↓ | NC Mean↑ | NC Med.↑ |
|---|---|---|---|---|---|---|
| 0% noise | 0.034 | 0.021 | 0.015 | 0.007 | 0.666 | 0.760 |
| 10% noise | 0.041 | 0.025 | 0.018 | 0.007 | 0.666 | 0.761 |
| 25% noise | 0.054 | 0.031 | 0.023 | 0.006 | 0.655 | 0.747 |

Table 14: **Effects of the cached memory token.**

| Method | Input Type | Acc.↓ | Comp.↓ | Overall↓ | Time↓ |
|---|---|---|---|---|---|
| **StreamVGGT** (W/ Causal Attention) | Full-sequence | 0.782 | 0.723 | 0.753 | 4.7 s |
| **StreamVGGT** (W/ Cached Token) | Streaming | 0.782 | 0.723 | 0.753 | 0.07 s |

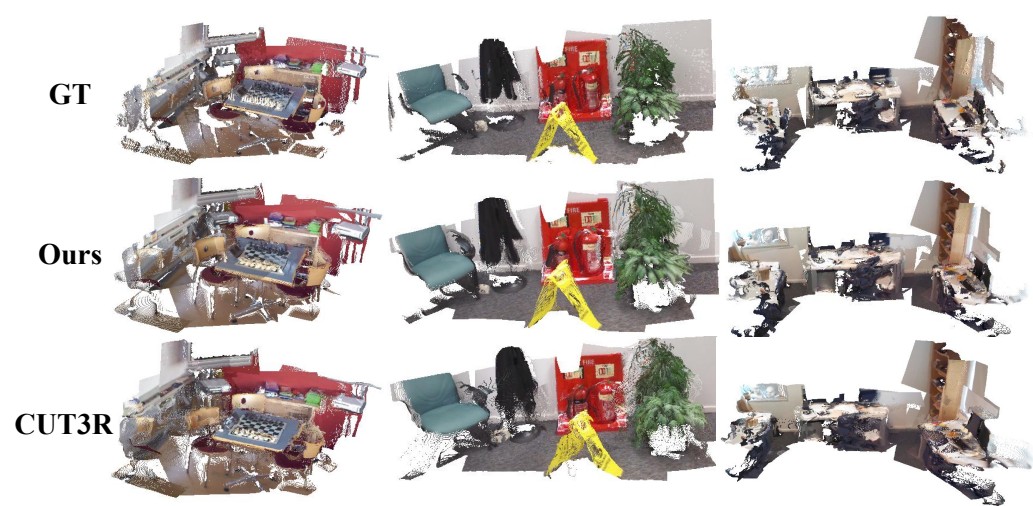

GT

Ours

CUT3R

Figure 7: **Visualization of the reconstruction results of StreamVGGT and CUT3R on 7-Scenes.**

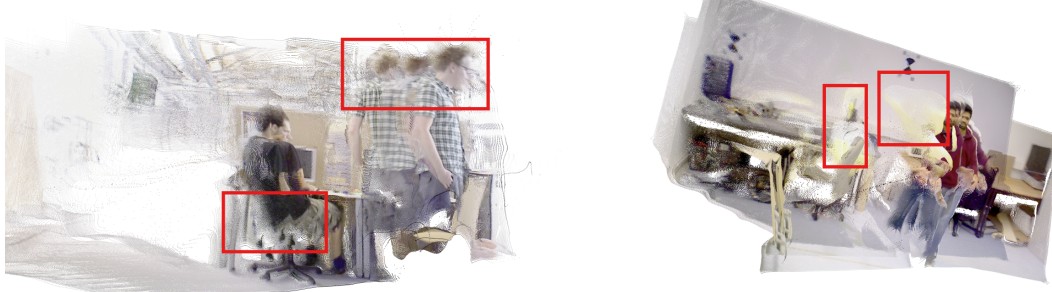

Figure 8: **Visualizations of failure cases.**

CUT3R (Wang et al., 2025b) on the 7-Scenes (Shotton et al., 2013) dataset. We compare our results of 3D reconstruction with CUT3R and the ground truth in Figure 7, demonstrating that our method achieves high-accuracy 3D reconstruction with fewer outliers and superior spatial consistency.

**Failure Cases.** We provide visualizations of some failure cases in Figure 8. These include instances of severe dynamic occlusion and aggressive ego-motion, which cause drift or localization failure.

**Visualizations of Pose Estimation.** We provide visualizations of pose estimation on ScanNet (Dai et al., 2017) in Figure 9 and Figure 10. The visualizations demonstrate that our StreamVGGT can accurately estimate camera poses across diverse scenarios and performs better than CUT3R (Wang et al., 2025b) when estimating the trajectories of camera pose.

**More Quantitative Results.** We provide a comprehensive comparison between StreamVGGT and a concurrent work STream3R (Lan et al., 2025) in Table 15, Table 16 and Table 17. Compared with

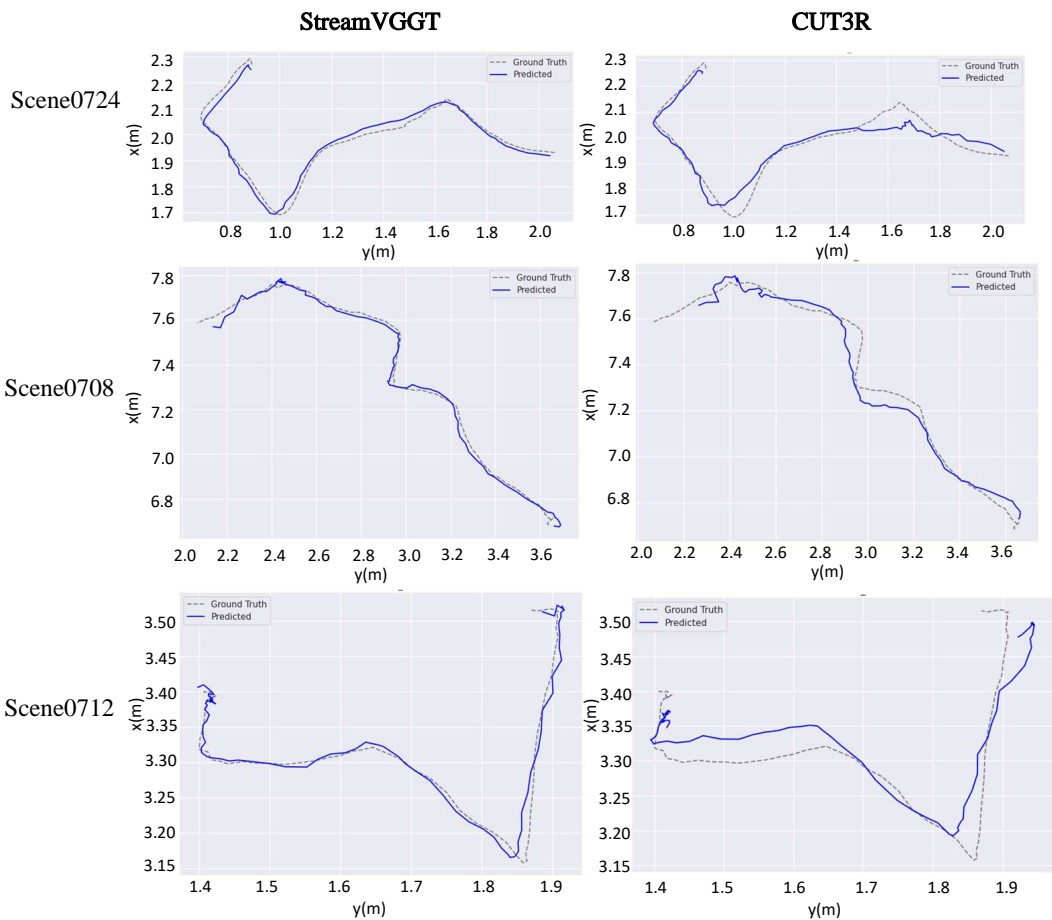

Figure 9: **Visualizations of the pose estimation of StreamVGGT and CUT3R on Scannet.**

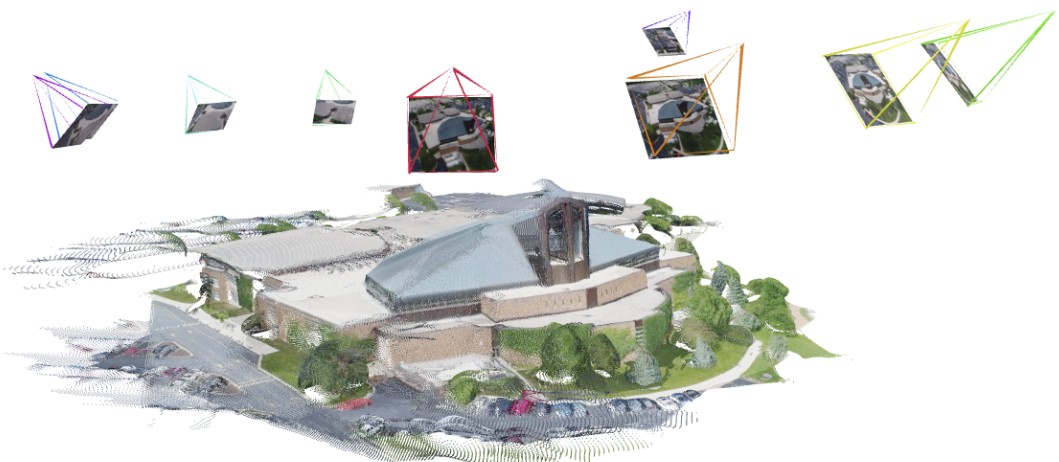

Figure 10: **Visualizations of point map and camera pose estimation of StreamVGGT.**

STream3R, which was trained on 29 datasets using 8 NVIDIA A100 GPUs over seven days, our StreamVGGT achieves comparable results under constrained training resources by using only 13 training datasets and 4 A800 GPUs over the same seven-day period. This fully demonstrates the effectiveness of our knowledge distillation strategy, which not only accelerates the training process but also significantly reduces the demand for training resources.

Table 15: **Quantitative 3D reconstruction results on 7-Scenes and NRGBD datasets.**

| | | 7 scenes | | | | | | NRGBD | | | | | |
| | | Acc↓ | | Comp↓ | | NC↑ | | Acc↓ | | Comp↓ | | NC↑ | |
| Method | Type | Mean | Med. | Mean | Med. | Mean | Med. | Mean | Med. | Mean | Med. | Mean | Med. |
|---|---|---|---|---|---|---|---|---|---|---|---|---|---|
| STream3R (Lan et al., 2025) | Streaming | **0.119** | 0.058 | **0.110** | **0.031** | 0.747 | 0.854 | **0.068** | **0.016** | **0.033** | **0.014** | **0.904** | 0.981 |
| **StreamVGGT** | Streaming | 0.129 | **0.056** | 0.115 | 0.038 | **0.751** | **0.865** | 0.084 | 0.044 | 0.074 | 0.041 | 0.861 | **0.986** |

Table 16: **Single-Frame Depth Evaluation.**

| | | Sintel | | Bonn | | KITTI | | NYU-v2 (Static) | |
| Method | Type | Abs Rel ↓ | $\delta<1.25$ ↑ | Abs Rel ↓ | $\delta<1.25$ ↑ | Abs Rel ↓ | $\delta<1.25$ ↑ | Abs Rel ↓ | $\delta<1.25$ ↑ |
|---|---|---|---|---|---|---|---|---|---|
| STream3R (Lan et al., 2025) | Streaming | **0.229** | **70.7** | 0.061 | 96.7 | **0.064** | **95.5** | 0.057 | 95.7 |
| **StreamVGGT** | Streaming | 0.254 | 68.5 | **0.052** | **97.1** | 0.072 | 94.7 | **0.055** | **95.9** |

Table 17: **Quantitative 4D reconstruction results on TUM-dynamics.**

| Method | Type | Acc Mean↓ | Acc Med.↓ | Comp Mean↓ | Comp Med.↓ | NC Mean↑ | NC Med.↑ |
|---|---|---|---|---|---|---|---|
| STream3R (Lan et al., 2025) | Streaming | 0.087 | 0.012 | **0.040** | **0.007** | 0.604 | 0.665 |
| **StreamVGGT** | Streaming | **0.085** | **0.011** | 0.058 | **0.007** | **0.617** | **0.690** |

## A.3 FURTHER DISCUSSIONS

**Limitations.** Although our cached token memory mechanism effectively retains historical frame information, this approach leads to a substantial increase in memory usage and computational overhead for long-term sequences as shown in Figure 2. As more frames are processed, the memory footprint grows rapidly due to the accumulation of cached tokens. This scalability issue poses a significant challenge for deploying the model on lightweight or mobile devices, where hardware resources are limited. Therefore, addressing memory efficiency while preserving accuracy remains a critical area for future optimization.

**Broader Impacts.** The StreamVGGT model offers significantly broader impacts across multiple industries, especially in the domain of low-latency 3D visual geometry reconstruction. By integrating temporal attention with a Cached Token Memory mechanism, StreamVGGT enables efficient incremental processing of multi-view images, ensuring low-latency scene updates with minimal latency while maintaining long-term spatial consistency. This innovative approach makes it a crucial technology for dynamic applications such as autonomous navigation, robotics, and immersive AR/VR experiences, where timely and accurate 3D scene understanding is essential.

