# OpenReview forum: "Streaming Visual Geometry Transformer"
_ICLR.cc/2026/Conference — ICLR 2026 Poster_

### Official Review · Reviewer_6AuX · 2025-10-27

**Soundness:** 3
**Presentation:** 3
**Contribution:** 3
**Rating:** 6
**Confidence:** 4

**Summary:**

The paper proposes an online, low-latency 3D reconstruction pipeline. It achieves this by (1) temporal causal attention, cross-attending to cached KV-tokens from previously observed frames, and (2) utilizing knowledge distillation during training by setting VGGT (a full SA-based reconstruction model) as the teacher to provide the GT for the causal student model.

**Strengths:**

1. The paper is well written and easy to follow. The problem is well motivated and explained.
2. The proposed causal attention is simple but effective, and performs well empirically.

**Weaknesses:**

1. In Table 6, the impact of KD is rather drastic, which makes the contribution of the causal attention questionable. When reporting results for w/o KD, is the 'true' GT used as supervision? The authors claim that KD helps reduce error accumulation, but this claim needs a more involved analysis.

**Questions:**

1. Did the authors explore strategies for constraining/truncating the memory usage for very long scenes? Perhaps simply incorporating a recency bias i.e. cacheing and cross-attending to only the most recent K scenes' tokens instead. An ablation over different K can be helpful.
2. Can the authors explain the results reported in Figure 6?

---

> ### Author Response · Authors · 2025-11-20
>
> Thanks for your thoughtful comments, questions and insightful suggestions.
> We provide detailed responses below.
>
> **W1-More explanation and discussion about the usage of KD**
>
> The use of causal attention is mainly for the ability to process streaming inputs online efficiently. This results in a faster speed and lower memory consumption while achieving performance comparable to VGGT.
>
> "w/o KD" in Table 6 of the main paper indeed refers to training the model using the 'true' GT as supervision. We agree that it is difficult to verify its effect on error reduction. We have modify relevant claims and the advantage of knowledge to this:
>
> - **Unified multi-task supervision reduces engineering burden.**
>
>  VGGT jointly predicts cameras, depth, point maps, and tracks, and our method reuses its loss design and supervises all heads with the teacher outputs as pseudo-GT. This yields coherent scaling/visibility conventions without per-dataset processing. Moreover, we do not need to relabel all the datasets for generating track GTs.
>
> - **Confidence acts as a regularizer, improving robustness and generalization**.
>
> Classic and modern KD papers show that the teacher soft targets encode implicit knowledge that regularize the student and mitigate label noise, often outperforming training solely on hard labels, especially when labels are scarce or noisy. VGGT not only outputs the target predictions but also gives confidence estimation. We utilize these confidences to supervise our model, which can improve the robustness and generalization.
>
>
> **Q1-Strategies for constraining the memory usage for very long scenes**
>
> Thanks for your advice. We therefore introduce two bounded-memory variants: (i) windowed streaming, which partitions the sequence into fixed-length chunks and then aligns the point clouds produced by the model for each chunk using the predicted camera extrinsics, ensuring that the peak memory stays below the window budget; and (ii) K-nearest-frames caching, where the current frame only attends to cached tokens from the most recent K frames. Table 1 shows both variants bound memory/latency on very long sequences while preserving accuracy, offering a practical efficiency–accuracy trade-off.
>
> ### Table 1 Comparison of pruning strategies for online inference on a 200-frame sequence.
> | Method          | Acc↓ (Mean/Med.)| Comp↓ (Mean/Med.)| NC↑ (Mean/Med.)| Inference Time (at Frame 200) | Peak Memory |
> | --------------- | ------- | -------- | ------- | -------- | ------- |
> | Spann3R | 0.225/0.176 | 0.070/0.015 | 0.538/0.557  | 221.054 ms  | 6.7 GB  |
> | CUT3R | 0.165/0.107 | 0.030/0.012 | 0.551/0.577  | 102.533 ms  | 7.6 GB  |
> | No Prunning | 0.058/0.035 | 0.026/0.010 | 0.632/0.708  | 733.083 ms  | 25.3 GB  |
> | Window-based (50 frames) | 0.054/0.030 | 0.023/0.009 | 0.636/0.712 | 219.454 ms  | 8.2 GB  |
> | Window-based (100 frames) | 0.062/0.033 | 0.032/0.015 | 0.633/0.710 | 392.634 ms  | 14.0 GB  |
> | K-nearest-frames (K=50) | 0.157/0.114 | 0.104/0.011 | 0.605/0.670 | 220.109 ms  | 8.4 GB  |
> | K-nearest-frames (K=100) | 0.054/0.032 | 0.034/0.019 | 0.626/0.702 | 392.634 ms  | 14.0 GB  |
>
> **Q2-Explanation about the results in Figure 6**
>
> We use Figure 6 to demonstrate that our approach maintains robust temporal consistency when handling long sequences.
> In each subplot of Figure 6, the horizontal axis indicates which frame of the long sequence the model is currently processing, and the vertical axis represents the per-frame metric computed by comparing the predicted point cloud of the model for this frame with the corresponding ground-truth point cloud.
> We can observe that, apart from the initial warm-up phase, the per-frame performance of our model remains very stable throughout the progressive processing of the long sequence.
>
> We hope the above response can help address your concerns.
> We are happy to answer any additional questions you may have.

---

### Official Review · Reviewer_adCa · 2025-10-30

**Soundness:** 3
**Presentation:** 3
**Contribution:** 3
**Rating:** 8
**Confidence:** 3

**Summary:**

This paper proposes StreamVGGT, which reconstructs 3D geometry from videos in an online manner. They do this by adopting a causal transformer architecture and caching the historical keys and values to use for each incremental new frame. They train StreamVGGT through knowledge distillation of VGGT, which is state-of-the-art, "offline", but slower. For 3D reconstruction, their approach increases inference speed relative to offline methods, but with a small performance reduction. They are competitive with (and sometimes slightly better than) other streaming methods.

**Strengths:**

1. This paper's approach of caching memory tokens for streaming reconstruction is important for efficiency, and it is useful for the field to see the results and comparisons they report. I think this makes the paper useful and worth publishing.
2. The paper is written clearly and provides a very readable explanation of their method and other methods.
3. StreamVGGT is competitive with other streaming methods, while adopting a significantly different approach. The additional experiments (inference speed, memory, distillation ablations) also serve as useful information.

**Weaknesses:**

1. It would be useful to explicitly compare the inference (and training, if possible, but understandably this may be harder) speed and memory consumption of their approach against other streaming reconstruction methods like CUT3R and Spann3R. I think this is crucial because the core argument of their paper is efficiency.
2. "StreamVGGT leverages the inherent sequential and causal nature of real-world video data, constraining the attention mechanism to past and current frames, thereby aligning with the causal structure observed in human perception." I agree that humans receive input frames in an online streaming manner. However, do we know if there is evidence that their representation of old views is not modified by (and so does not "attend to") new views, like in KV caching? One counterpoint is that our stored experiences and memories are modified by new information. It is possible that humans use some hybrid: KV-caching where stored keys/values are allowed to be modified. At a few points in this paper, the paper implicitly suggests that StreamVGGT is more human-like than full self-attention approaches. I think this requires more evidence, or otherwise hedge the claim.

**Questions:**

1. To what extent is the inference speedup and memory reduction (Figure 2) due to memory token caching versus FlashAttention-2?
2. I think it would be helpful to discuss: what are the different methods (e.g., StreamVGGT, CUT3R) good at and where do they fail? How is this linked to their method (architecture, training strategy, training data, etc.)?

---

> ### Author Response · Authors · 2025-11-20
>
> Thanks for your thoughtful comments, questions and insightful suggestions.
> We provide detailed responses below.
>
> **W1-More comparison on training and inference time and memory consumption with CUT3R and Spann3R**
>
> We now provide more comparison on training and inference consumption with CUT3R and Spann3R. As shown in Table 1, our method exhibits a moderate increase in inference time and memory usage when processing longer sequences. This is mainly because the number of cached memory tokens grows with the sequence length. Consequently, compared with CUT3R and Spann3R, our method requires more computation and memory for very long sequences. However, we want to point out that there is a trade-off between spatial memory storage and performance, and StreamVGGT shows a good one especially for not very long sequences.
>
> ### Table 1 The comparison of inference-time and memory consumption for the online setting. Acc indicates Accuracy. Lower acc is better.
>
> | Method          | 1 frame | 5 frames | 10 frames | 20 frames | 30 frames | 40 frames |
> | --------------- | ------- | -------- | --------- | --------- | --------- | --------- |
> | Spann3R  | 0.039 Acc / 500 ms / 5.3 GB   | 0.038 Acc / 87 ms / 5.4 GB  | 0.046 Acc / 82 ms /  5.5 GB  |  0.036 Acc / 120 ms / 5.5 GB   | 0.043 Acc / 192 ms / 5.6 GB   | 0.068 Acc / 125 ms / 5.7 GB   |
> | CUT3R | 0.035 Acc / 101 ms / 3.4 GB   | 0.038 Acc / 102 ms / 3.5 GB   | 0.039 Acc / 99 ms / 3.6 GB     | 0.031 Acc / 101 ms / 3.8 GB     | 0.027 Acc / 102 ms / 4.0 GB     | 0.023 Acc / 102 ms / 4.2 GB     |
> | **StreamVGGT (Ours)**  | 0.028 Acc / 63 ms / 2.1 GB   | 0.024 Acc / 88 ms / 2.7 GB    | 0.024 Acc/ 120 ms / 3.2 GB | 0.021 Acc / 152 ms / 4.3 GB     | 0.021 Acc / 184 ms / 5.5 GB     | 0.021 Acc / 216 ms / 6.6 GB     |
>
> Still, StreamVGGT faces scalability challenge for very long sequences as shown in Table 2. To address this, we introduce two bounded-memory processing strategies: (i) windowed streaming, which partitions the sequence into fixed-length chunks and then aligns the point clouds produced for each chunk using the predicted camera extrinsics, so that peak memory stays within a predefined budget; and (ii) K-nearest-frames caching, where each frame attends only to tokens from the most recent K frames. As demonstrated in Table 2, both approaches effectively bound memory usage and latency during inference on extended videos, while maintaining high accuracy. This offers a practical trade-off that effectively resolves computational concerns for long-sequence handling without significant degradation in results.
>
> ### Table 2 Comparison of pruning strategies for online inference on a 200-frame sequence.
> | Method          | Acc↓ (Mean/Med.)| Comp↓ (Mean/Med.)| NC↑ (Mean/Med.)| Inference Time (at Frame 200) | Peak Memory |
> | --------------- | ------- | -------- | ------- | -------- | ------- |
> | Spann3R | 0.225/0.176 | 0.070/0.015 | 0.538/0.557  | 221.054 ms ms  | 6.7 GB  |
> | CUT3R | 0.165/0.107 | 0.030/0.012 | 0.551/0.577  | 102.533 ms  | 7.6 GB  |
> | No Prunning | 0.058/0.035 | 0.026/0.010 | 0.632/0.708  | 733.083 ms  | 25.3 GB  |
> | Window-based (50 frames) | 0.054/0.030 | 0.023/0.009 | 0.636/0.712 | 219.454 ms  | 8.2 GB  |
> | Window-based (100 frames) | 0.062/0.033 | 0.032/0.015 | 0.633/0.710 | 392.634 ms  | 14.0 GB  |
> | K-nearest-frames (K=50) | 0.157/0.114 | 0.104/0.011 | 0.605/0.670 | 220.109 ms  | 8.4 GB  |
> | K-nearest-frames (K=100) | 0.054/0.032 | 0.034/0.019 | 0.626/0.702 | 392.634 ms  | 14.0 GB  |
>
>
> Regarding training cost, Table 3 further highlights that our approach is significantly more efficient than CUT3R. Our model can be trained on 4× A800 GPUs for only 7 days. Moreover, we use only 13 datasets for training (less than half of CUT3R’s dataset size). This demonstrates that our method not only delivers higher accuracy, but also offers substantial savings in training time, computational resources, and dataset scale.
>
> ### Table 3 A comparison of the training strategies of StreamVGGT, CUT3R and Spann3R.
>
> | Dimension | **StreamVGGT (Ours)**  | **CUT3R** | **Spann3R** |
> | --------- | ---------------------- | --------- | ----------- |
> | **Training strategy**           | 13 datasets covering static/dynamic & synthetic/real data + multi-resolution inputs.  **Knowledge** **distillation** from the offline VGGT teacher. The training runs on 4 A800 GPUs 7 days. | Multi-stage curriculum, multi-resolution + 32 training datasets. Random ray-map replacement to learn “see-through” inference. The training runs on 8 A100 GPUs over a month. |Curriculum with random 5-frame sampling on 6 training datasets; scheduled memory growth/shrink. The training runs on 8 V100 GPUs around 10 days.|

---

> > ### Comment · Reviewer_adCa · 2025-11-23
> >
> > The authors' response addressed my comments. In particular, I like that they present some methods for reducing their inference time (rebuttal Table 2), which I think is important. Maybe that is worth putting into the paper (or supplementary). Overall, I think this paper is good and I maintain my positive score.

---

> > > ### Author Response · Authors · 2025-11-27
> > >
> > > Dear Reviewer adCa,
> > >
> > > Thanks for your time and efforts. Your insightful review and valuable suggestions have significantly improved the clarity and quality of our work.
> > >
> > > Best regards, Authors of Submission 4557

---

> > > ### Author Response · Authors · 2025-11-28
> > >
> > > Dear Reviewer adCa,
> > >
> > > Thank you once again for your time and thoughtful review. Your constructive suggestions are helpful, and we have carefully followed them and incorporated the corresponding results into the revised paper.
> > >
> > > Best regards, Authors of Submission 4557

---

> ### Author Response · Authors · 2025-11-20
>
> **W2-More discussion on the human-like casual structure**
>
> Thanks for the nice suggestion. We agree that you have a very good point.
> We mainly think processing observations in an online streaming manner is more consistent with human cognition. We have modified relevant sentences to avoid confusion.
>
> **Q1-The effect of memory token caching and FlashAttention-2 on inference speedup and memory reduction**
>
> Thanks for the suggestion. To evaluate the effect of memory token caching and FlashAttention-2 on inference speedup and memory reduction, we now provide the ablation experiment in Table 4. The results clearly demonstrate that memory token caching reduces inference time and the FlashAttention-2 reduces the memory overhead.
>
> ### Table 4 Inference time and memory comparison for the online setting.
> | Method          | Inference Time (at Frame 5) | Peak Memory |
> | --------------- | ------- | -------- |
> | W/O FlashAttn & Cache | 1135.860 ms  | 5.4 GB  |
> | W/ FlashAttn |  850.699 ms  |  2.3 GB  |
> | W/ FlashAttn & Cache | 88.193 ms  | 2.7 GB   |
>
> **Q2-More thorough comparison with existing methods**
>
> ### Table 5. Comparison of Strengths, Failure Cases, and Underlying Causes among StreamVGGT, CUT3R, and Spann3R.
>
> As summarized in Table 5, we explicitly compare what each method (StreamVGGT, CUT3R, Spann3R) is good at and where it fails, and further link these behaviors to their architectural choices, training strategies, and training data. Our analysis highlights that StreamVGGT benefits from causal modeling and knowledge distillation, enabling better cross-dataset generalization and more efficient training while maintaining high accuracy in the online setting. CUT3R trades long training time for strong reconstruction ability, and Spann3R favors lightweight memory at the cost of long-sequence stability.
>
> | **Dimension** | **StreamVGGT (Ours)** | **CUT3R** | **Spann3R** |
> |--------------|-----------------------|-----------|-------------|
> | **Overall positioning** | Real-time **streaming 4D reconstruction** with strong temporal consistency and generalization. | General-purpose **continual-update 3D perception**; strong surface completion via ray queries. | Lightweight **online point-map fusion** optimized for memory-limited settings. |
> | **What it is good at** | • Strong **temporal consistency** via causal attention + cached memory. • Excellent **cross-dataset generalization** due to VGGT-based KD. • Robust on long sequences with minimal drift. | • Exceptional **completion of unseen regions** via ray-map replacement. • Stable pose updates using persistent state tokens. • Good on sparse-view scenes. | • Very **fast** and **low-memory** inference. • Performs well on short, mostly static indoor scenes. |
> | **Where it fails** | • Inference-time & memory grow with sequence length. • Very fast camera motion may exceed causal receptive field. | • Long sequences cause drift due to limited global modeling. •  Training requires significantly longer time due to multi-stage curriculum and recurrent optimization, making scaling expensive. | • Rapid drift on long sequences due to bounded memory. • Poor generalization to outdoor or dynamic scenes. |
> | **Architectural cause** | • **Implicit cached memory token** grows with sequence length. • Pure causal transformer lacks future-frame context. | • **Persistent state tokens** encode history but cannot maintain global consistency over long ranges. | • **Fixed-size spatial memory** limits long-term context; aggressive pruning removes useful tokens. |
> | **Training strategy cause** | • **Knowledge distillation** reduces training cost while maintaining good generalization. • Multi-resolution, synthetic/real, static/dynamic training improves generality. | • **Multi-stage curriculum** + ray-map replacement improves completion but requires substantial training resources. | • **Short 5-frame curriculum** strengthens local consistency but weakens long-sequence modeling. |
> | **Training data cause** | • 13 curated datasets → balanced indoor/outdoor, dynamic/static; strong generalization. | • 32+ datasets → high diversity aids surface reconstruction but leads to longer training time. | • Mostly indoor datasets → limited robustness in dynamic/outdoor environments. |
>
> We hope the above response can help address your concerns. We are happy to answer any additional questions you may have.

---

### Official Review · Reviewer_XfbV · 2025-10-31

**Soundness:** 4
**Presentation:** 4
**Contribution:** 4
**Rating:** 6
**Confidence:** 5

**Summary:**

This paper proposes StreamVGGT, a causal transformer architecture designed for real-time streaming 3D visual geometry reconstruction. Inspired by the success of large language models with causal attention, the authors integrate temporal attention and a cached token memory mechanism to enable efficient, incremental 3D scene updates without needing to reprocess full sequences. Extensive experiments on multiple 3D reconstruction benchmarks demonstrate that Stream3R achieves competitive performance.

**Strengths:**

1. This demonstrates that using cache token technology can significantly improve VGGT's inference performance with long video inputs.
2. The technical details are clear and easy to follow.

**Weaknesses:**

1. The authors’ demo video showcases the model’s capability to handle scenes with significant motion. It would be helpful if the paper included more analysis or discussion regarding these dynamic scenarios.
2. I don't quite follow the motivation behind using knowledge distillation (KD). First, why not directly train the model on real-world data? Is it because the dataset size and quality are insufficient compared to VGGT, leading to suboptimal results? Second, since VGGT cannot be trained on long video sequences due to the computational cost of attention, wouldn't this limitation of the teacher model affect StreamVGGT’s performance on long sequences—especially for tasks like pose estimation?
3. Figure 2 showcases the time efficiency advantage over the vanilla VGGT. However, since the authors emphasize the streaming nature of their approach, it would be more convincing if they could also compare the overall reconstruction time with other similar methods, such as CUT3R.

**Questions:**

1. Since there are already many similar works (e.g., Stream3R, Lan et al.) that also adopt caching techniques, it would be beneficial if the authors could include a comparison with these approaches in the paper.
2. Adding a pose estimation visualization and a comparison with CUT3R would make the paper more complete.

---

> ### Author Response · Authors · 2025-11-20
>
> Thanks for your thoughtful comments, questions and insightful suggestions.
> We provide detailed responses below.
>
> **W1-More analysis on dynamic scenarios**
>
> Thanks for your nice advice.
> Apart from the qualitative results on dynamic scenes shown in our demo video, we also provide quantitative metrics on dynamic datasets in Appendix A.1 of the main paper. We compare our model with VGGT and CUT3R on the TUM-dynamics dataset. Because our model does not rely on any additional input requirements or priors, it fully supports dynamic scenes. As shown in the appendix, our method outperforms other online approaches by a significant margin, surpassing CUT3R and achieving results comparable to VGGT.
>
> **W2-The motivation behind using knowledge distillation**
>
> Thanks for your question.
> We use knowledge distillation to train our model mainly for the following reasons:
>
> - **Unified multi-task supervision reduces engineering burden.**
>
>  VGGT jointly predicts cameras, depth, point maps, and tracks, and our method reuses its loss design and supervises all heads with the teacher outputs as pseudo-GT. This yields coherent scaling/visibility conventions without per-dataset processing. Moreover, we do not need to relabel all the datasets for generating track GTs.
>
> - **Confidence acts as a regularizer, improving robustness and generalization**.
>
> Classic and modern KD papers show that the teacher soft targets encode implicit knowledge that regularize the student and mitigate label noise, often outperforming training solely on hard labels, especially when labels are scarce or noisy. VGGT not only outputs the target predictions but also gives confidence estimation. We utilize these confidences to supervise our model, which can improve the robustness and generalization.
>
> Our experiments in Table 6 of our main paper have shown that with the help of knowledge distillation, even though our training dataset size and quality are insufficient compared to VGGT, we are still able to train a high-performing model with a relatively small amount of GPU hours. Although VGGT cannot be trained on very long sequences, we believe that the performance and training efficiency gains brought by knowledge distillation are more important. This is well demonstrated by the fact that we surpass existing online methods on the camera pose estimation task. Moreover, thanks to our causal design, we can continue training on long sequences using ground-truth supervision after a well-trained model is obtained. However, this additional refinement does not conflict with our usage of knowledge distillation during the main training stage.
>
> **W3-Time efficiency comparison with more baselines**
>
> Table 1 indicates that the latency and memory usage of StreamVGGT increase moderately with sequence length owing to the growth of cached tokens. Nevertheless, despite higher cost than CUT3R/Spann3R on very long sequences, the overhead remains acceptable and is compensated by consistently better accuracy.
>
> ### Table 1 The comparison of inference-time and memory consumption for the online setting. Acc indicates accuracy. Lower acc is better.
>
> | Method          | 1 frame | 5 frames | 10 frames | 20 frames | 30 frames | 40 frames |
> | --------------- | ------- | -------- | --------- | --------- | --------- | --------- |
> | Spann3R  | 0.039 Acc / 500 ms / 5.3 GB   | 0.038 Acc / 87 ms / 5.4 GB  | 0.046 Acc / 82 ms /  5.5 GB  |  0.036 Acc / 120 ms / 5.5 GB   | 0.043 Acc / 192 ms / 5.6 GB   | 0.068 Acc / 125 ms / 5.7 GB   |
> | CUT3R | 0.035 Acc / 101 ms / 3.4 GB   | 0.038 Acc / 102 ms / 3.5 GB   | 0.039 Acc / 99 ms / 3.6 GB     | 0.031 Acc / 101 ms / 3.8 GB     | 0.027 Acc / 102 ms / 4.0 GB     | 0.023 Acc / 102 ms / 4.2 GB     |
> | **StreamVGGT (Ours)**  | 0.028 Acc / 63 ms / 2.1 GB   | 0.024 Acc / 88 ms / 2.7 GB    | 0.024 Acc/ 120 ms / 3.2 GB | 0.021 Acc / 152 ms / 4.3 GB     | 0.021 Acc / 184 ms / 5.5 GB     | 0.021 Acc / 216 ms / 6.6 GB     |

---

> ### Author Response · Authors · 2025-11-20
>
> **Q1-Comparison with STream3R**
>
> Thank you for the suggestion. We will add more discussions. We provide a comprehensive comparison between StreamVGGT and STream3R in Tables 2-4. Compared with STream3R, which was trained on 29 datasets using 8 NVIDIA A100 GPUs over seven days, our StreamVGGT achieves comparable results under constrained training resources by using only 13 training datasets and 4 A800 GPUs over the same seven-day period. This fully demonstrates the effectiveness of our knowledge distillation strategy, which not only accelerates the training process but also significantly reduces the demand for training resources. Also note that Stream3R was only released on arXiv on Aug. 14th, 2025 and should be considered as concurrent work as ours.
>
> ### Table 2 Quantitative 3D reconstruction results on 7-Scenes and NRGBD datasets.
> |Method|Type|7 scenes Acc↓ (Mean/Med.)|7 scenes Comp↓ (Mean/Med.)|7 scenes NC↑ (Mean/Med.)|NRGBD Acc↓ (Mean/Med.)|NRGBD Comp↓ (Mean/Med.)|NRGBD NC↑ (Mean/Med.)|
> |:--:|:--:|:--:|:--:|:--:|:--:|:--:|:--:|
> |STream3R|Streaming|**0.119**/0.058|**0.110**/**0.031**|0.747/0.854|**0.068**/**0.016**|**0.033**/**0.014**|**0.904**/0.981|
> |**StreamVGGT (Ours)**|Streaming|0.129/**0.056**|0.115/0.038|**0.751**/**0.865**|0.084/0.044|0.074/0.041|0.861/**0.986**|
>
> ### Table 3 Single-Frame Depth Evaluation.
> |**Method**|**Type**|Sintel Abs Rel↓/δ<1.25↑|Bonn Abs Rel↓/δ<1.25↑|KITTI Abs Rel↓/δ<1.25↑|NYU-v2 Abs Rel↓/δ<1.25↑|
> |:--:|:--:|:--:|:--:|:--:|:--:|
> |STream3R|Streaming|**0.229**/**70.7**|0.061/96.7|**0.064**/**95.5**|0.057/95.7|
> |**StreamVGGT (Ours)**|Streaming|0.254/68.5|**0.052**/**97.1**|0.072/94.7|**0.055**/**95.9**|
>
> ### Table 4 Quantitative 4D reconstruction results on TUM-dynamics dataset.
>
> |**Method**|**Type**|Acc Mean↓|Acc Med.↓|Comp Mean↓|Comp Med.↓|NC Mean↑|NC Med.↑|
> |:--:|:--:|:--:|:--:|:--:|:--:|:--:|:--:|
> |STream3R|Streaming|0.087|0.012|**0.040**|**0.007**|0.604|0.665|
> |**Ours**|Streaming|**0.085**|**0.011**|0.058|**0.007**|**0.617**|**0.690**|
>
> **Q2-1-Pose estimation visualization**
>
> We provide the visualizations of pose estimation with a comparison with CUT3R in Appendix A.2 of the updated main paper. The visualizations demonstrate that our StreamVGGT can accurately estimate camera poses across diverse scenarios, including those with dynamic environments.
>
>
> We hope the above response can help address your concerns.
> We are happy to answer any additional questions you may have.

---

### Official Review · Reviewer_KPdZ · 2025-11-07

**Soundness:** 3
**Presentation:** 4
**Contribution:** 3
**Rating:** 6
**Confidence:** 4

**Summary:**

This paper introduces StreamVGGT, a transformer architecture for streaming 3D visual geometry reconstruction. The model draws inspiration from autoregressive language models and replaces global self-attention with temporal causal attention, enabling incremental 3D reconstruction with cached token memory. StreamVGGT supports online inference while maintaining spatial and temporal consistency. The approach achieves comparable reconstruction and depth estimation performance to offline methods (e.g., VGGT) while substantially improving inference speed on long sequences.

**Strengths:**

* The paper addresses a practical problem: enabling low-latency 3D reconstruction suitable for real-time streaming systems.
* The temporal causal attention and cached memory are conceptually simple, computationally efficient, and align well with online mapping needs.
* The experimental evaluation is extensive, covering multiple datasets and comparing against both dense view and streaming baselines.
* The writing is clear, structured, and easy to follow.
* The paper demonstrates significant latency reduction (e.g., 10× faster inference for long sequences) while maintaining similar accuracy compared to VGGT.

**Weaknesses:**

* The issue of error accumulation over long sequences is not thoroughly analyzed. No explicit mechanisms (e.g., drift correction) are introduced beyond distillation, and the paper does not present accuracy trends across different sequence lengths compared to VGGT.
* The scalability and memory-growth behavior of cached tokens for very long video sequences remains unclear.
* The connection to causal or autoregressive modeling is somewhat superficial: there is no explicit probabilistic formulation or next-frame prediction, only causal masking.
* The paper does not explore qualitative failure cases, such as dynamic occlusions or fast ego-motion, which are critical for practical deployment.

**Questions:**

* How does the model handle error accumulation or drift over long sequences (e.g., hundreds or thousands of frames)?
* Does the cached token memory ever saturate or require pruning? If so, how is this managed without degrading accuracy?
* Can the causal model generalize beyond static datasets (e.g., driving, dynamic human motion)?
* How robust is StreamVGGT to missing or noisy frames, which are common in real streaming data?

---

> ### Author Response · Authors · 2025-11-20
>
> Thanks for your thoughtful comments, questions and insightful suggestions.
> We provide detailed responses below.
>
> **W1 & Q1-Error accumulation over long sequences**
>
> We have provided an initial analysis of the long-sequence behavior of our method in Appendix A.1 of our paper. We now add experiments on hundreds of frames, providing accuracy trends and a head-to-head comparison with VGGT and online method CUT3R in Table 1a-1c, showing comparable accuracy of our StreamVGGT in long sequences. These results demonstrate that error accumulation is an unavoidable issue for online methods, but our approach stores sufficient historical information via cached tokens to mitigate this problem. Nevertheless, there remains room for improvement. We believe that pruning stored tokens or introducing a window-based mechanism could be promising directions for further reducing error accumulation.
>
> ### Table 1a The accuracy trends of StreamVGGT across different sequence lengths.
>
> |Sequence Length|Acc Mean↓|Acc Med.↓|Comp Mean↓|Comp Med.↓|NC Mean↑|NC Med.↑|
> |:--:|:--:|:--:|:--:|:--:|:--:|:--:|
> |1 Frame|0.028|0.015|0.032|0.021|0.827|0.914|
> |5 Frames |0.024|0.013|0.026|0.011|0.748|0.866|
> |10 Frames |0.024|0.012|0.023|0.009|0.692|0.794|
> |20 Frames|0.021|0.008|0.020|0.005|0.646|0.723|
> |30 Frames|0.021|0.008|0.016|0.004|0.627|0.700|
> |40 Frames|0.021|0.009|0.015|0.004|0.632|0.707|
> |70 Frames|0.020|0.008|0.013|0.003|0.610|0.671|
> |100 Frames|0.017|0.006|0.010|0.003|0.607|0.666|
> |150 Frames|0.014|0.004|0.007|0.002|0.589|0.638|
> |200 Frames|0.012|0.004|0.006|0.001|0.582|0.626|
> |250 Frames|0.016|0.008|0.008|0.002|0.574|0.619|
>
> ### Table 1b The accuracy trends of VGGT across different sequence lengths.
>
> |Sequence Length|Acc Mean↓|Acc Med.↓|Comp Mean↓|Comp Med.↓|NC Mean↑|NC Med.↑|
> |:--:|:--:|:--:|:--:|:--:|:--:|:--:|
> |1 Frame |0.029|0.017|0.035|0.024|0.836|0.914|
> |5 Frames|0.024|0.014|0.034|0.024|0.780|0.892|
> |10 Frames|0.024|0.013|0.034|0.024|0.761|0.880|
> |20 Frames|0.021|0.009|0.033|0.023|0.701|0.803|
> |30 Frames|0.019|0.006|0.029|0.020|0.650|0.732|
> |40 Frames|0.015|0.005|0.025|0.017|0.630|0.703|
> |70 Frames|0.009|0.003|0.018|0.010|0.612|0.673|
> |100 Frames|0.008|0.002|0.015|0.006|0.607|0.667|
> |150 Frames|0.007|0.002|0.011|0.004|0.591|0.642|
> |200 Frames|0.006|0.002|0.008|0.002|0.583|0.628|
> |250 Frames|0.005|0.002|0.008|0.001|0.575|0.620|
>
> ### Table 1c The accuracy trends of CUT3R across different sequence lengths.
> |Sequence Length|Acc Mean↓|Acc Med.↓|Comp Mean↓|Comp Med.↓|NC Mean↑|NC Med.↑|
> |:--:|:--:|:--:|:--:|:--:|:--:|:--:|
> |1 Frame |0.035|0.014|0.039|0.016|0.796|0.891|
> |5 Frames|0.038|0.015|0.029|0.008|0.691|0.788|
> |10 Frames|0.039|0.019|0.027|0.006|0.650|0.731|
> |20 Frames|0.031|0.011|0.021|0.004|0.619|0.685|
> |30 Frames|0.027|0.008|0.019|0.004|0.560|0.654|
> |40 Frames|0.023|0.006|0.019|0.003|0.591|0.640|
> |70 Frames|0.029|0.010|0.021|0.003|0.580|0.624|
> |100 Frames|0.031|0.014|0.019|0.003|0.576|0.618|
> |150 Frames|0.030|0.012|0.015|0.003|0.562|0.595|
> |200 Frames|0.030|0.011|0.016|0.002|0.555|0.584|
> |250 Frames|0.038|0.013|0.013|0.002|0.553|0.580|

---

> ### Author Response · Authors · 2025-11-20
>
> **W2 & Q2-Scalability and memory-growth behavior of cached tokens for very long video sequences**
>
> Our method achieves higher efficiency than VGGT precisely because it leverages cached tokens and exhibits stronger scaling capability. However, cached tokens still grow linearly with sequence length, and compared with other online methods, our approach continues to face certain bottlenecks as shown in Table 2. We therefore introduce two bounded-memory variants: (i) windowed streaming, which partitions the sequence into fixed-length chunks and aligns the point clouds for each chunk using the predicted camera extrinsics, so that the peak memory stays below the window budget; and (ii) K-nearest-frames caching, where the current frame only attends to cached tokens from the most recent K frames. As shown in Table 3, both controls keep memory/latency bounded on very long videos while maintaining near-unchanged accuracy (or only a slight drop), providing a practical trade-off without degrading results.
>
>
> ### Table 2 The comparison of inference-time and memory consumption for the online setting. Acc indicates Accuracy. Lower acc is better.
>
> | Method          | 1 frame | 5 frames | 10 frames | 20 frames | 30 frames | 40 frames | 70 frames | 100 frames| 150 frames | 200 frames |
> | --------------- | ------- | -------- | --------- | --------- | --------- | --------- | --------- | --------- | --------- | --------- |
> | VGGT | 0.029 Acc / 63 ms / 2.1 GB   | 0.024 Acc / 184 ms / 3.7 GB  | 0.024 Acc / 386 ms /  5.6 GB  |  0.021 Acc / 870 ms / 7.5 GB   | 0.019 Acc / 1320 ms / 9.4 GB   | 0.015 Acc / 2089 ms / 11.4 GB   |   0.009 Acc / 5343 ms / 17.8 GB   | 0.008 Acc / 9502 ms / 24.8 GB   |  0.007 Acc / 19070 ms / 35.9 GB   | -- / OOM   |
> | Spann3R  | 0.039 Acc / 500 ms / 5.3 GB   | 0.038 Acc / 87 ms / 5.4 GB  | 0.046 Acc / 82 ms /  5.5 GB  |  0.036 Acc / 120 ms / 5.5 GB   | 0.043 Acc / 192 ms / 5.6 GB   | 0.068 Acc / 125 ms / 5.7 GB   |  0.195 Acc / 164 ms / 5.9 GB   | 0.252 Acc / 92 ms / 6.1 GB   |  0.200 Acc / 85 ms / 6.4 GB   | 0.165 Acc / 221 ms / 6.7 GB   |
> | CUT3R | 0.035 Acc / 101 ms / 3.4 GB   | 0.038 Acc / 102 ms / 3.5 GB   | 0.039 Acc / 99 ms / 3.6 GB     | 0.031 Acc / 101 ms / 3.8 GB     | 0.027 Acc / 102 ms / 4.0 GB     | 0.023 Acc / 102 ms / 4.2 GB     | 0.029 Acc / 102 ms / 4.9 GB     | 0.031 Acc / 103 ms / 5.5 GB     |  0.030 Acc / 104 ms / 6.6 GB     | 0.030 Acc / 102 ms / 7.6 GB     |
> | **StreamVGGT (Ours)**  | 0.28 Acc / 63 ms / 2.1 GB   | 0.024 Acc / 88 ms / 2.7 GB    | 0.024 Acc / 120 ms / 3.2 GB | 0.021 Acc / 152 ms / 4.3 GB     | 0.021 Acc / 184 ms / 5.5 GB     | 0.021 Acc / 216 ms / 6.6 GB     | 0.020 Acc / 282 ms / 10.1 GB     | 0.017 Acc / 386 ms / 13.7 GB     | 0.014 Acc / 562 ms / 19.7 GB     | 0.012 Acc / 733 ms / 23.2 GB     |
>
> ### Table 3 Comparison of pruning strategies for online inference on a 200-frame sequence.
> | Method          | Acc↓ (Mean/Med.)| Comp↓ (Mean/Med.)| NC↑ (Mean/Med.)| Inference Time (at Frame 200) | Peak Memory |
> | --------------- | ------- | -------- | ------- | -------- | ------- |
> | Spann3R | 0.225/0.176 | 0.070/0.015 | 0.538/0.557  | 221.054 ms ms  | 6.7 GB  |
> | CUT3R | 0.165/0.107 | 0.030/0.012 | 0.551/0.577  | 102.533 ms  | 7.6 GB  |
> | No Prunning | 0.058/0.035 | 0.026/0.010 | 0.632/0.708  | 733.083 ms  | 25.3 GB  |
> | Window-based (50 frames) | 0.054/0.030 | 0.023/0.009 | 0.636/0.712 | 219.454 ms  | 8.2 GB  |
> | Window-based (100 frames) | 0.062/0.033 | 0.032/0.015 | 0.633/0.710 | 392.634 ms  | 14.0 GB  |
> | K-nearest-frames (K=50) | 0.157/0.114 | 0.104/0.011 | 0.605/0.670 | 220.109 ms  | 8.4 GB  |
> | K-nearest-frames (K=100) | 0.054/0.032 | 0.034/0.019 | 0.626/0.702 | 392.634 ms  | 14.0 GB  |
>
> **W3-Connection to causal or autoregressive modeling**
>
> Sorry for the confusion. We want to clarify that, the term "causal" in our paper is used in contrast to global interaction in VGGT.
> The inference for a frame only replys on previous frames but not future ones, which is causal.
> Yet, it does not rely on explicit probabilistic modeling or other techniques to predict/generate the next frame based on previous inputs.
> In this sense, we think our model is more like "causal" but not "autoregressive".
>
> **W4-More analysis on failure cases**
>
> We appreciate your valuable feedback regarding the discussion of failure cases. Following this suggestion, we have expanded Appendix A.2 with new visualizations that delve into specific failure modes. These include cases of severe dynamic occlusion and aggressive ego-motion, which cause drift or complete localization failure.

---

> ### Author Response · Authors · 2025-11-20
>
> **Q3-Generalization to dynamic datasets**
>
> Our model can generalize beyond static datasets because it does not rely on any additional input requirements or priors. We compared our model with VGGT, CUT3R on TUM-dynamics dataset in Appendix A.1 of our main paper. The results demonstrate that when handling dynamic scenes, our method outperforms other online approaches (like CUT3R) by a large margin and achieves results comparable to VGGT.
>
> **Q4-Robustness to missing or noisy frames**
>
> The Appendix A.1 of our main paper has shown that StreamVGGT remains robust to missing frames, maintaining high accuracy under intermittent gaps. For noisy frames, we further corrupt inputs by randomly masking regions to simulate sensor noise/occlusion. Table 4 shows reconstruction remains correct, which proves the robustness of our StreamVGGT to noisy frames.
>
> ### Table 4 The robustness performance under noisy frames.
>
> | Method          | Acc↓ (Mean/Med.)| Comp↓ (Mean/Med.)| NC↑ (Mean/Med.)|
> | --------------- | ------- | -------- | ------- |
> | 0% noise | 0.034 / 0.021 | 0.015 /  0.007  | 0.666 / 0.760 |
> | 10% noise | 0.041 / 0.025 | 0.018 / 0.007 |  0.666 / 0.761 |
> | 25% noise |  0.054 / 0.031 |  0.023 / 0.006 | 0.655 / 0.747  |
>
> We hope the above response can help address your concerns.
> We are happy to answer any additional questions you may have.

---

### Author Response · Authors · 2025-12-03

Dear PCs, SACs, ACs, and all reviewers,

We first want to express our gratitude for your great efforts to our paper. Your feedback greatly improved the clarity and completeness of our submission. We appreciate the reviewers for highlighting the following strengths of our work:

- A causal, low-latency streaming architecture that enables online 4D reconstruction with competitive accuracy.
- An effective cached token memory design that preserves historical context and stabilizes long-sequence streaming reconstruction.
- A VGGT-based knowledge distillation strategy that retains multi-task supervision while substantially reducing engineering overhead and training cost.

We have also addressed the main concerns raised by reviewers:

- We clarified the motivation and role of our knowledge distillation strategy, and revised claims to emphasize its concrete benefits: unified multi-task supervision without per-dataset processing, and confidence-based soft targets that regularize training for better robustness and generalization.
- We expanded long-sequence evaluations (hundreds of frames) and provided accuracy trends and comparisons with VGGT and CUT3R, showing stable performance.
- We acknowledged linear memory growth with sequence length and introduced two bounded-memory variants (windowed streaming and K-nearest-frame caching), with new results demonstrating bounded latency/memory on 200-frame sequences while maintaining accuracy.
- We added thorough efficiency analysis and ablations, showing that cached token memory speeds up inference and FlashAttention-2 reduces peak memory.
- We strengthened comparisons to STream3R, including training-cost evidence showing StreamVGGT reaches comparable performance using fewer datasets and fewer GPU resources than STream3R.
- We expanded failure-case and robustness discussions, and provided quantitative results on dynamic and noisy/missing-frame settings, confirming strong generalization to complex scenes.

Finally, we would like to thank all the reviewers recognizing our contribution to feed-forward streaming reconstruction. We believe our causal design, KD-driven training strategy and cached token memory mechanism would provide a practical accuracy–efficiency balance for online 4D reconstruction.

Thanks again to all of you for your time and efforts.

Best regards,

Authors of Submission 4557

---

### Public Comment · ~Zihan_Lin4 · 2026-04-16
**Citation of Sintel dataset is wrong**

The paper cites *Sintel: A machine learning framework to extract insights from signals* as sintel dataset in the paper, but this paper is about time series, which has no relationship with sintel dataset.

---

### Meta-Review · Area_Chair_vZZx · 2026-01-07

**Summary:**

StreamVGGT proposes a causal transformer for online 3D reconstruction from video. It uses temporal causal attention and caches historical keys and values as an implicit memory so each new frame updates the scene without reprocessing the full sequence. It trains the causal model by distilling from the offline VGGT teacher and adopts efficient attention operators (e.g., FlashAttention) at inference. Main strengths are clear speed gains with strong accuracy and clean efficiency ablations separating caching vs FlashAttention. However, cached memory still grows with sequence length (bounded variants help but are a trade-off), and novelty over prior streaming caching methods is limited.

**Reviewer Concerns:**

Addressed: long-sequence behavior (hundreds of frames), memory growth and scalability (windowed streaming and K-nearest caching), clearer KD motivation, more runtime and memory comparisons, and ablations showing what caching vs FlashAttention changes.

Still outstanding: linear-growth is not fully fixed (bounded options trade accuracy), and the core idea is close to existing streaming/caching work.

**Reviewer Scores:**

adCa: stays at 8 (they explicitly say the rebuttal addressed their comments and they keep the positive score).

KPdZ: likely 6 to higher after long-sequence trends, bounded-memory variants, and failure-case discussion.

XfbV: likely 6 to higher after KD clarification and stronger baseline comparisons.

6AuX: likely stay at 6.

---

### Decision · Program_Chairs · 2026-01-26

Accept (Poster)